# Federated Multi-armed Bandits with Efficient Bit-Level Communications

**Haoran Zhang**
Nanjing University
602023710004@smail.nju.edu.cn

**Yang Xu**
Nanjing University
yang.xu@smail.nju.edu.cn

**Xuchuang Wang**
University of Massachusetts Amherst
xuchuangw@gmail.com

**Hao-Xu Chen**
Nanjing University
2718522637@qq.com

**Hao Qiu**
Università degli Studi di Milano
hao.qiu@unimi.it

**Lin Yang**[*]
Nanjing University
linyang@nju.edu.cn

**Yang Gao**
Nanjing University
gaoy@nju.edu.cn

## Abstract

In this work, we study the federated multi-armed bandit (FMAB) problem, where a set of agents collaboratively aim to minimize cumulative regret. Unlike traditional centralized bandit models, agents in FMAB settings are connected via a communication graph and cannot share data freely due to bandwidth limitations or privacy constraints. This raises a fundamental challenge: how to achieve optimal learning performance under stringent communication budgets. We propose a novel communication-efficient algorithm containing two points: one for eliminating suboptimal arms through early and frequent communication of key decisions, and the other for refining global estimates using incremental epoch, quantized, and differentially transmitted statistics. *Incremental Epoch-based Successive Elimination Algorithm* (`EpoInc-SE`) is presented by carefully balancing communication frequency and precision of global estimates. Theoretically, we derive tight upper bounds on both individual cumulative regret and group regret, and prove that our method asymptotically matches the lower bound of regret in federated settings. Experimental results on synthetic data validate the effectiveness of `EpoInc-SE` in various settings and under heterogeneous feedback.

## 1 Introduction

The stochastic multi-armed bandit (MAB) problem is a classic and influential research topic in the field of sequential decision-making [Bandits, Lattimore and Szepesvári, 2020]. In this model, a decision-maker must select an arm from a set of actions at each time step to pull and receive a random reward. The core objective is to maximize cumulative rewards, or equivalently, minimize cumulative regret, which is defined as the reward lost compared to always selecting the optimal arm. Due to its powerful modeling capabilities and wide applicability, the MAB framework has been successfully applied in numerous domains, such as clinical trials [Wang, 1991], online advertising recommendations [Tang et al., 2015] and resource allocation [Lattimore et al., 2015].

In recent years, with the development of distributed systems and increasing awareness of data privacy, traditional single-agent MAB models have gradually extended to multi-agent collaborative scenarios,

---
[*]Corresponding author

39th Conference on Neural Information Processing Systems (NeurIPS 2025).

Table 1: Comparison of existing FMAB algorithms with our proposed method `EpoInc-SE`.

| Algorithm | Central Server | Hete-/Homo- | Individual Regret | Group Regret | Bit-Com |
|---|---|---|---|---|---|
| `Gossip_UCB` [Zhu et al., 2021] | ✗ | Hete- | $O(\sum_{i:\Delta_i>0} N\Delta_i^{-1}\log T)$ | $O(\sum_{i:\Delta_i>0} N^2\Delta_i^{-1}\log T)$ | $O(T\log T)$ |
| `Dis_UCB` [Zhu and Liu, 2023] | ✗ | Hete- | $O(\sum_{i:\Delta_i>0} N_{\min}^{-1}\Delta_i^{-1}\log T)^1$ | $O(\sum_{i:\Delta_i>0} N N_{\min}^{-1}\Delta_i^{-1}\log T)$ | $O(T\log T)$ |
| `UCB-TCOM` [Wang and Yang, 2023] | ✗ | Homo- | $O(\sum_{i:\Delta_i>0} N^{-1}\Delta_i^{-1}\log T)$ | $O(\sum_{i:\Delta_i>0} \Delta_i^{-1}\log T)$ | $\tilde{O}(\log T)$ |
| `Fed2-UCB` [Shi and Shen, 2021] | ✓ | Homo- | $O(\sum_{i:\Delta_i>0} N^{-1}\Delta_i^{-1}\log T)$ | $O(\sum_{i:\Delta_i>0} \Delta_i^{-1}\log T)$ | $O(T\log T)$ |
| `EpoInc-SE` (our work) | ✗ | Hete- | $O(\sum_{i:\Delta_i>0} N^{-1}\Delta_i^{-1}\log T)$ | $O(\sum_{i:\Delta_i>0} \Delta_i^{-1}\log T)$ | $O(\sqrt{\log T})$ |
| Regret lower bound | ✗ | Hete- | $\Omega(\sum_{i:\Delta_i>0} N^{-1}\Delta_i^{-1}\log T)$ | $\Omega(\sum_{i:\Delta_i>0} \Delta_i^{-1}\log T)$ | – |

[1] $N_{\min}$ denotes the smallest number of neighbors for any agent, including the agent itself.

where the Federated Multi-Armed Bandit (FMAB) problem has garnered increasing attention [Li et al., 2024, Boursier and Perchet, 2024]. In large-scale sensor networks, environmental constraints and cost considerations often prevent full physical connectivity or the deployment of a central server, naturally leading to the need for distributed information processing and computation [Xiao et al., 2005, Golovin et al., 2010]. Moreover, online service platforms with many servers often store large volumes of data locally. To protect user privacy, centralizing this data for computation and training is impractical, where federated learning is a useful and crucial approach to complete the two tasks Ellison and Fudenberg [1995], Sankararaman et al. [2019].

In FMAB problems, the primary objective is to minimize the total regret accumulated across all agents, referred to as *group regret* in this paper. This metric has been widely studied in prior works [Kalathil et al., 2014, Shi and Shen, 2021, Xu and Klabjan, 2024]. However, in many practical scenarios, the performance of individual agents also plays a critical role, as it can become the bottleneck that limits overall system effectiveness. In network optimization literature [Srikant and Ying, 2014], the max-min fairness metric—maximize the minimal individual reward—is widely used to measure a system's fairness. Another important metric in FMAB problems is *communication bit* incurred by all agents. In real-world scenarios, communication is often expensive and establishing peer-to-peer (P2P) connectivity may be impractical. To evaluate this cost, the number of *communication rounds* is a commonly used metric [Cho et al., 2020, Li and Song, 2022]. However, a single round may involve transmitting multiple messages, which is misleading for evaluating an algorithm. Therefore, a more precise and informative measure is the total number of *communication bits*, which has been adopted in recent studies [Wang et al., 2020, Agarwal et al., 2022].

**Related Works.** The most relevant prior works on FMAB have been summarized in Table 1. Among them, `Gossip_UCB` [Zhu et al., 2021] and `Dis_UCB` [Zhu and Liu, 2023] are the two works most related to ours, which are all heterogeneous FMAB without a central server. However, they could not obtain the optimal results compared with `EpoInc-SE` (our work). `Fed2-UCB` [Shi and Shen, 2021] and `UCB-TCOM` [Wang and Yang, 2023] are two typical Homgeneous FMAB algorithms that use a central server and fully distributed communication, respectively. Meanwhile, they all achieve the optimal result and `UCB-TCOM` [Wang and Yang, 2023] also optimizes the communication rounds. The two algorithms provide classic frameworks for bandit learning. Hence, we apply a consensus estimator on them and take them as the baseline in Section 5. Several works have investigated the regret lower bound in federated bandit settings [Xu and Klabjan, 2023, Wang et al., 2020]. Specifically, they establish group regret lower bounds for heterogeneous and homogeneous FMAB problems, respectively. In addition, Xu and Klabjan [2023] shows that, in the absence of information exchange with neighbors, an agent in a federated setting can suffer linear regret $O(T)$. For bit communication techniques, there are not too many works on heterogeneous FMAB. But there are also some interesting works about bit communication in other directions [McMahan et al., 2017, Casteigts et al., 2019, Wang et al., 2019, Boursier and Perchet, 2019, Shi et al., 2021]. Wang et al. [2019] and Shi et al. [2021] reduced the communication rounds of the algorithm under a homogeneous setting. These works [McMahan et al., 2017, Boursier and Perchet, 2019] considered communication bits for the first time in bandit problems. The truncation of messages is proposed in Casteigts et al. [2019], Shi et al. [2021]. Our communication optimization method combines the advantages of works above.

**Contribution.** Addressing the aforementioned challenges, this paper proposes a novel federated multi-armed bandit algorithm designed for fully distributed communication networks. It enables

agents to effectively solve the global MAB problem and simultaneously optimize individual and group regrets by communicating solely with their neighbors. Distinct from previous work, our main contributions and innovations include:

**(1) Optimization of Communication Efficiency:** `EpoInc-SE` is the first work considering the communication cost in heterogeneous federated bandit learning. By employing epoch-based exploration to reduce communication rounds and designing an adaptive difference communication mechanism, we significantly reduce the communication rounds and the number of effective information bits per communication round. Additionally, we introduce a two-phase communication strategy to differentiate the transmission priority of different types of information, thereby enhancing communication efficiency while preserving regret performance.

**(2) Near-Optimal Regret Performance:** We propose the first lower bound of heterogeneous federated bandit $\Omega(\sum_i^N \Delta_i^{-1} N^{-1} \log T)$ with the dependence of the number of agent $N$, which matches the upper bound of `EpoInc-SE` in terms of agent number $N$, reward gap $\Delta_i$ and time horizon $T$, thereby proving the optimality of our algorithm design and analysis.

**(3) Efficient Online Consensus Estimator:** We design and introduce an epoch-based consensus estimation subroutine `EBCES` that allows each agent to unbiasedly estimate the global mean of arms under the condition of fully distributed communication and buffering broadcast. This is crucial for resolving heterogeneity and communication optimization.

The remainder of this paper is organized as follows: Section 2 details the problem model, including the communication model, the federated multi-armed bandit model, and performance metrics. Section 3 introduces our proposed communication-efficient algorithm design, covering its core ideas, epoch-based exploration strategy, arm elimination mechanism, adaptive difference communication scheme, and consensus estimation policy. Section 4 provides the theoretical analysis of the algorithm, including proofs for regret upper and lower bounds. Section 5 provides simulations to validate the theoretical analysis. Finally, Section 6 concludes the paper.

## 2    Problem Formulation

**Communication Model.** Consider a fully distributed network, which works only via finite paths among agents, i.e., without a central server. The data exchanged between agents, called a message, is a string of binary numbers. The network is a (fully) connected graph, defined as $\mathcal{G} = (\mathcal{N}, \mathcal{E}, \mathcal{A})$ [Huang et al., 2022]. $\mathcal{N}$ is an agent set, where agents are labeled by $\{1, \ldots, N\}$. The $\mathcal{E} \subset \mathcal{N} \times \mathcal{N}$ denotes the set of edges in the network. $\mathcal{A} = [a_{a,b}]_{(a,b) \in \mathcal{N} \times \mathcal{N}}$ is an adjacency matrix, used to determine whether two agents are connected. For agent $j$, define its neighborhood as $\mathcal{N}_j$. To meet computing needs, an additional matrix $\boldsymbol{W} = [\omega_{a,b}]_{(a,b) \in \mathcal{N} \times \mathcal{N}}$ describes the weight of agents in their neighborhoods. The definition of $\boldsymbol{W}$ is $\boldsymbol{W} = \boldsymbol{I} - \beta\mathcal{L}$, where $\mathcal{L}$ is the Laplacian matrix and $\beta \in (0, 1/N]$ is the coefficient that reflects whether this agent believes its neighbors. If $\beta = 0$, we have $\boldsymbol{W} = \boldsymbol{I}$, which means that the agent itself occupies all the weight. If $\beta = 1/N$, it indicates that the agent assigns equal trust to all of its neighbors.

**System Model.** There are $N$ agents repeatedly making decisions from a fixed arm set $\mathcal{K} = \{1, \ldots, K\}$ over a time horizon $T$. At each time step $t$, agent $j$ selects an arm $A_j(t)$ and receives a reward $X_{i,j}(t)$ if $A_j(t) = i$. The reward $X_{i,j}(t)$ is drawn from a $\frac{1}{2}$-sub-Gaussian distribution with an unknown (local) mean $\mu_{i,j} \in [0, 1]$. These rewards serve as real-time feedback to guide agents in learning the expected rewards of the arms. In federated settings, the reward $X_{i,j}$ is a local observation that agent $j$ samples from arm $i$. For arm $A_j(t) = i$, there also exists a global reward $X_i(t)$ that is contributed by all agents' observations, i.e., $X_{A_j(t)}(t) = X_i(t) := \sum_{j=1}^N X_{i,j}(t)$. The mean of $X_i$ is denoted by the global mean $\mu_i := \frac{1}{N}\sum_{j=1}^N \mu_{i,j} \in [0, 1]$. Without loss of generality, let $i^\star := \arg\max_i \mu_i$ denote the unique optimal arm with the highest global mean among all arms. Define the suboptimality gap for each arm $i$ as $\Delta_i := \mu_{i^\star} - \mu_i$. In this work, we consider a heterogeneous setting where the local reward mean of arm $i$ may differ across agents, i.e., $\mu_{i,j_1} \neq \mu_{i,j_2}$, for $j_1 \neq j_2$. There is no collision in this setting, implying that all agents can obtain rewards without degradation, even if they pull the same arm.

**Individual regret.** The individual regret of agent $j$ is defined as the cumulative regret incurred if all agents are to follow agent $j$'s decision. This metric is crucial in federated learning, where the

| Symbol/Term | Definition |
|---|---|
| $\mathcal{K}$ | The arm set |
| $\mathcal{N}$ | The agent set |
| $A_j(t)$ | The decision of agent $j$ at time slot $t$, $A_j(t) \in \mathcal{K}$ |
| $X_{i,j}(t)$ | The local reward of arm $i$ sampled by agent $j$ at time slot $t$, $X_{i,j}(t) \in [0,1]$ |
| $\mu_{i,j}$ | The local mean of arm $i$ for agent $j$, $\mu_{i,j} \in [0,1]$ |
| $X_i(t)$ | The global reward of arm $i$, $X_i(t) = \sum_{j=1}^{N} X_{i,j}(t)$ |
| $\mu_i$ | The global mean of arm $i$, $\mu_i = \sum_{j=1}^{N} \mu_{i,j}$ |
| $i^\star$ | The optimal arm with the largest global mean, $i^\star \in \arg\max_i \mu_i$ |
| $\Delta_i$ | The gap between arm $i$ and the optimal arm, $\Delta_i = \mu_{i^\star} - \mu_i$ |

Table 2: Symbols of the multi-agent bandit problem

objective is to identify the arm with the highest global mean reward. Since relying solely on local rewards can be misleading, individual regret provides a more meaningful measure of an agent's contribution to the global goal. The formal definition is given as follows,

$$\mathbb{E}[R_j(T)] := T\mu_{i^\star} - \sum_{t=1}^{T} \mathbb{E}[X_{A_j(t)}(t)]. \tag{1}$$

**Group regret.** The group regret is defined as the cumulative regret caused by all agents, which is also seen as a main performance metric of the algorithm. The optimal performance of an algorithm is that all agents pull the optimal arm during the time horizon $T$. Hence, for a distributed algorithm, the expected group regret of the entire system is defined as

$$\mathbb{E}[R(T)] := \sum_{j=1}^{N} \mathbb{E}[R_j(T)]. \tag{2}$$

**Communication cost.** Besides two regrets, the communication cost is also an important performance metric to evaluate a federated bandit algorithm. Two types of communication metrics are commonly used in previous works: communication rounds and communication bits. While the communication costs are primarily measured by the number of communication rounds, e.g., Tao et al. [2019], Yang et al. [2024], the number of communication bits is a more practical and realistic metric that accurately reflects the real-world cost of data transmission. In this paper, we define the communication cost in bit level, denoted $C(T)$ as the total number of bits consumed from communicating all messages among all agents. In this work, we assume that any message (a number of bits) can be broadcast within a single time slot. For any messages, we do not consider the identifiers, and there is no limitation on the message size.

## 3   `EpoInc-SE`: A Communication-Efficient Algorithm Design

### 3.1   Key Ideas and Algorithm Structure

To achieve optimal regret bounds under limited communication, the core challenge lies in ensuring that the exchanged information is just sufficient to support accurate consensus estimation. Due to the heterogeneity in rewards, these local estimates often deviate from the global mean, making communication with neighbors necessary. This creates a fundamental trade-off: sharing highly precise and real-time information provides limited improvement to the convergence of consensus estimation but results in excessive communication overhead. In Boursier and Perchet [2019], Wang et al. [2019], information quantization and epoch-based exploration techniques were proposed to control the size of the message adaptively. However, these approaches are typically developed for homogeneous bandits with centralized structures, such as having a leader agent directly connected to all followers, and thus do not apply to settings with heterogeneity and fully distributed graphs.

In this work, we extend the classical successive elimination algorithm to a fully distributed setting, achieving an improved regret bound of $O(\sum_{i:\Delta_i>0} N^{-1}\Delta_i^{-1} \log T)$. Building on this, we further optimize the communication cost at the bit level, which comprises two key components: the number

of communication rounds (optimized in Section 3.2) and the message size (optimized in Section 3.3). The first component leverages a batched sampling strategy to reduce communication rounds. In distributed bandit settings, each agent must sample arms sufficiently to ensure accurate local estimates. Rather than communicating after every pull, agents accumulate multiple samples before sharing with neighbors. This reduces communication frequency and ensures that shared information is more reliable, particularly as early-stage observations tend to be noisy. Epoch-based exploration thus offers a principled trade-off between statistical efficiency and communication cost, making the approach scalable in large, decentralized systems. The second component for optimizing communication complexity reduces the bit-width of the transmitted messages. Three kinds of statistics need to be optimized, including the indices of arms, the sample count of each arm, and the global estimate. The indexes of arms consume only $O(\log K)$ bits, which has no room for optimization. The sample count is implicit information considering the algorithm's synchronization. Hence, we improve the message size of global estimates from $O(\log T)$ to $O(1)$ bits. To realize this task, we propose a differential encoding approach where agents share only the quantized change between consecutive estimates instead of the full values.

To further reduce communication costs while maintaining regret performance, we categorize shared information into two types based on their impact on the algorithm's efficiency. The first type includes indexes of eliminated arms, which are broadcast immediately to accelerate arm elimination and improve regret. The second type of shared information includes global estimates of arm means, which are tightly coupled with the sampling process. Sharing them too frequently does not directly contribute to faster convergence, but can significantly increase communication overhead. Therefore, this type of information should be buffered and communicated less frequently. Based on this insight, our algorithm divides communication into two distinct phases. **Communication Phase I** is responsible for sharing the indexes of eliminated arms. This phase operates at a high frequency to ensure the timely elimination of suboptimal arms across agents. In contrast, **Communication Phase II** handles the exchange of global estimates. The frequency of this phase is intentionally kept low to limit communication costs. This design is especially effective when updates are frequent but minor, avoiding redundant communication.

Below, we introduce the details of the algorithm design. We start from the epoch-based exploration paradigm in Section 3.2, and then introduce three key modules of the algorithm design: an adaptive difference communication in Section 3.3, a consensus estimation policy in Section 3.4 and an arm elimination policy in Section 3.5.

## 3.2 The Epoch-based Exploration

In this section, we primarily introduce the epoch-based exploration strategy. To reduce the number of communication rounds overhead in federated learning, a buffering scheme that compresses communication rounds by delaying broadcasts is proposed. The main challenge is that excessive compression may cause newly collected local data to dominate the consensus estimation, which affects the contribution of agents' local observations in the global estimate (introduced in Section 3.4). Therefore, we carefully make a trade-off between minimizing communication rounds and maintaining the accuracy of the consensus estimation.

Next, we present an incremental epoch-based exploration where later epochs contain increasingly more time slots. Without loss of generality, we concentrate the discussion on a specific agent $j$. The time horizon $T$ is partitioned into a series of subprocesses, labeled by $r_j = 1, 2, \ldots$. In epoch $r_j$, agents maintain a candidate set $\mathcal{S}_j^r$ to decide which arm to pull. The initial set is $\mathcal{S}_j^0 = \mathcal{K}$. In epoch $r_j$, all arms in the candidate set $\mathcal{S}_j^r$ will be sampled $p^{r_j} = a(r_j + 1)$ times uniformly, where $a$ is a hyper-parameter that determines the size of each epoch. Therefore, the number of epochs is at most $O(\sqrt{T})$ for the time horizon $T$. The time slot at the beginning of the epoch $r_j$ is denoted by $t_{r_j}$, and the local empirical estimate on the arm $i$ during epoch $r_j$ is defined as follows

$$\bar{X}_{i,j}^r := \frac{\sum_{t_{r_j}}^{t_{r_j+1}-1} X_{i,j}(t)\mathbb{I}\{A_j(t) = i\}}{a(r_j + 1)}. \tag{3}$$

**Algorithm 1:** Epoch-based Successive Elimination Algorithm (`EpoInc-SE`)(for agent $j$)

---

**Input:** Weight matrix of graph $\boldsymbol{W}$, time horizon $T$, arm set $\mathcal{K}$, diameter $D$ and integer $a$

**Output:** $t = 0, r_j = 0, U_{i,j}^r = 1, \mathcal{S}_j = \mathcal{K}, \bar{X}_{i,j} = 0, \mathcal{B}_j = \varnothing$ for all arms $i$

1   Let agent $j$ pull each arm $a$ times and receive a sequence of rewards $\{X_{i,j}(t)\}_{t=1}^{t=a}$;

2   $\tilde{\mu}_{i,j}^r \leftarrow \operatorname{average}(X_{i,j}), \tilde{\mu}_{i,j}^r \leftarrow \bar{X}_{i,j}^r$;

3   **for** $r_j = 1, 2, \ldots$ **do**

4     $i^{\max} \leftarrow \arg\max_{i \in \mathcal{S}_j} \tilde{\mu}_{i,j}^r$;

5     **for** $p^{r_j} \in \{1, \ldots, a(r_j + 1)\}$ **do**

6       **for** $i \in \mathcal{S}_j$ **do**

7         Pull arm $i$ and obtain the reward $X_{i,j}$;

8         $\bar{X}_{i,j} \leftarrow (\bar{X}_{i,j}(p^{r_j} - 1) + X_{i,j})/p^{r_j}$;

      *// Communication phase I*

9       Receive $\mathcal{B}_{j'}$ and $\hat{r}_i$ from agent $j$'s neighborhood $\mathcal{N}_j$;

10      **if** $\mathcal{B}_j \geq 1$ **then**

11        Send $\mathcal{B}_j, \hat{r}_i$ to all agent $j$'s neighbors;

12       Update the new candidate set $\mathcal{B}_j \leftarrow \mathcal{B}_j \cup \left( \bigcup_{j' \in \mathcal{N}_j} B_{j'} \right)$;

    *// Communication phase II*

13     Update the global estimates via the adaptive difference communication (Algorithm 2);

14     Update $U_{i,j}^r$ via equation (6) ;

15     **for** $i \in \mathcal{S}_j$ **do**

16       **if** $\tilde{\mu}_{i,j}^{r-1} \leq \tilde{\mu}_{i^{\max},j}^{r-1} - 2U_{i,j}^{r-1}$ **then**

17         $\hat{r}_i \leftarrow r_j + \left\lceil \frac{|\mathcal{S}_j|D}{a(r_j + 1)} \right\rceil, \mathcal{B}_j \leftarrow \mathcal{B}_j \cup \{i\}$;

18     **for** *each arm $i$ in $\mathcal{B}_j$ whose $r_j \geq \hat{r}_i$* **do**

19       **if** $|\mathcal{S}_j| > 1$ **then** $\mathcal{S}_j \leftarrow \mathcal{S}_j \backslash \{i\}, \mathcal{B}_j \leftarrow \mathcal{B}_j \backslash \{i\}$; **else** $\mathcal{S}_j \leftarrow \mathcal{S}_j$

---

### 3.3 Adaptive Difference Communication

In this section, we optimize the communication cost by minimizing the message size. As outlined in Section 3.1, global estimates occupy the main channel resources and they consume $O(\log T)$ bits per message. Hence, we concentrate on optimizing the global estimate $\tilde{\mu}_{i,j}^r$ (introduced in Section 3.4).

A native intuition is to send brief information instead of precise global estimates. In the early stages of consensus estimation, high precision is unnecessary—initial rounds only require estimates accurate to a single decimal place. This motivates the use of information quantization, where the quantized global estimate at epoch $r_j$ is denoted as $\bar{\mu}_{i,j}^r$. The quantization is adaptive: larger messages yield higher accuracy. Specifically, the estimate in epoch $r_j$ is represented using $\lceil 1 + \log_2 r_j \rceil$ bits.

To further reduce the message size, another approach is based on sharing differential information rather than full local estimates. Instead of transmitting quantized statistics at each round, agents communicate only the differences between their current and previous values, i.e., quantization error $\delta_{i,j}^r := \bar{\mu}_{i,j}^r - \bar{\mu}_{i,j}^{r-1}$. This technique leverages the temporal smoothness of learning processes, significantly reducing the communication overhead while maintaining estimation accuracy.

Due to that $\bar{\mu}_{i,j}^r$ has $\lceil 1 + \log_2 r_j \rceil$ bits and $\bar{\mu}_{i,j}^r$ is a quantization version of $\tilde{\mu}_{i,j}^r$, the error between them is bounded by

$$|\bar{\mu}_{i,j}^r - \tilde{\mu}_{i,j}^r| \leq \frac{1}{2^{\lceil 1 + \log_2 r_j \rceil}} \leq \frac{1}{2r_j} \overset{r_j \geq 2}{\leq} \frac{1}{r_j + 2} \tag{4}$$

**Remark 1.** *Wang et al. [2019] has proved that $O(\log T)$ bits are sufficient to distinguish the optimal arm from the arm set $\mathcal{K}$, considering that the gap $\Delta_i$ may be small enough. The global estimates contribute a large number of bits to the communication cost. By exploiting the quantized variable $\bar{\mu}_{i,j}^r$, we reduce the message size to $O(\sqrt{\log T})$. By using differential communication, we can only*

**Algorithm 2:** Epoch-based consensus estimation subroutine (EBCES) (for agent $j$)

**Input:** The local reward $X_{i,j}$, the candidate set $\mathcal{S}_{j'}$, the epoch $r_j$ and the weight matrix $W = [\omega_{j,j'}]_{N \times N}$

**Output:** The latest estimate $\tilde{\mu}_{i,j}$ and elimination arm set $\mathcal{B}_j$

1 **if** $|\mathcal{S}_j| > 1$ **then**
2   $\quad$ Communicate the global error $\delta_{i,j}^{r-1}$ with its neighbors ;      *// Communication phase II*
3   $\quad$ Receive the global estimate $\delta_{i,j'}^{r-1}, j' \in \mathcal{N}_j$ from agent $j$'s neighbors;
4 **for** $j' \in \mathcal{N}_j$ **do**
5   $\quad$ $\bar{\mu}_{i,j'}^{r-1} \leftarrow \bar{\mu}_{i,j'}^{r-2} + \delta_{i,j'}^{r-1}$;
6 Update the global estimate $\mu_{i,j}^r$ via equation (5);
7 $\bar{\mu}_{i,j}^r \leftarrow \texttt{ceil}(\tilde{\mu}_{i,j}^r)$;
8 $\delta_{i,j}^r \leftarrow \bar{\mu}_{i,j}^r - \bar{\mu}_{i,j}^{r-1}$;

---

*use $O(1)$ bits per message. It is also the first work that obtains optimal message size in heterogeneous federated bandit learning.*

## 3.4 Consensus Estimation Policy

The heterogeneous reward in this problem motivates agents to learn the global means of arms and cooperate with their neighborhoods. Therefore, an epoch-based consensus estimation subroutine (EBCES), exploiting epoch-based exploration (Section 3.2) and differential information quantization (Section 3.3), is proposed to be integrated into `EpoInc-SE`.

The core idea of EBCES is to synchronize information exchange among neighbors and effectively utilize their shared data to achieve an unbiased global estimate. As discussed in Section 3.3, agents exchange $\delta_{i,j}^r$ instead of $\tilde{\mu}_{i,j}^r$ and reduce the information bits to $O(1)$ order. In the consensus estimation process, agents combine differential information $\delta_{i,j}^r$ with the historical quantized variable $\bar{\mu}_{i,j}^{r-1}$ to compute the latest global estimates from neighbors (Line 5, Algorithm 2). On the basis of the information exchange above, a fair estimation mechanism is presented in which samples from all agents are used equally to estimate the global mean.

To obtain global estimates, agent $j$ combines its neighbors' historical data and its own average empirical reward $\bar{X}_{i,j}^r$. Up to the end of epoch $r_j$, agent $j$ collects $\tilde{\mu}_{i,j'}^{r-1}, j' \in \mathcal{N}_j$ and updates its global estimate as follows:

$$\tilde{\mu}_{i,j}^r = (1 - \sigma_i(r_j)) \sum_{j' \in \mathcal{N}_j} \omega_{j,j'} \tilde{\mu}_{i,j'}^{r-1} + \sigma_i(r_j) \bar{X}_{i,j}^r, \tag{5}$$

where $\sigma_i(\cdot)$ is the weight that adjusts the contribution of the information in the global estimate $\tilde{\mu}_{i,j}^r$.

## 3.5 Arm Elimination Policy

In this section, we introduce an elimination arm set $\mathcal{B}_j^r$ that stores the indexes of arms identified as suboptimal. At the beginning of each epoch, each agent compares the global estimates of all arms and selects the one with the highest estimated reward as the baseline (Line 4, Algorithm 1). Then, each arm will be compared with arm $i^{\max}$, which is considered sub-optimal and added to the elimination set $\mathcal{B}_j$ if it falls below the baseline by a margin that is related to the confidence radius.

Denote the sample count of agent $j$ on arm $i$ until epoch $r_j$ by $\tau_{i,j}^r = a(r_j + 1)(r_j + 2)/2$. The total sample count on arm $i$ is $\tau_i^r = N\tau_{i,j}^r$ because of synchronous sampling. Define $\tilde{\mu}_{i,j}^r$ as the global estimate of agent $j$ on arm $i$. According to Hoeffding's inequality, there exists a radius of confidence interval $U_{i,j}^r$ such that $\tilde{\mu}_{i,j}^r \in [\mu_i - U_{i,j}^r, \mu_i + U_{i,j}^r]$, with

$$U_{i,j}^r(\delta) := \underbrace{\sqrt{\frac{\log \delta^{-1}}{aN(r_j + 1)(r_j + 2)}}}_{(a)} + \underbrace{\frac{2C}{(r_j + 2)(1 - \lambda_2)}}_{(b)} + \underbrace{\frac{1}{r_j + 2}}_{(c)}, \tag{6}$$

where $\delta$ specifies the violation probability. Terms (a) and (b) are from Lemmas 2 and 3. Term (c) is caused by information quantization (introduced in Section 3.3). The detailed proof of the confidence interval (6) is proposed in Appendix D.1. Arm $i$ is identified as sub-optimal if it satisfies

$$\tilde{\mu}_{i,j}^r + U_{i,j}^r \leq \tilde{\mu}_{i^{\max},j}^r - U_{i,j}^r. \tag{7}$$

To help agent $j$ determine whether the information in $\mathcal{B}_j^r$ has been acknowledged by all agents, we introduce epoch label $\hat{r}_i$, which denotes the latest epoch by which all agents are guaranteed to know that arm $i$ is suboptimal. Then, all agents agree to eliminate it from the candidate set at epoch $\hat{r}_i$,

$$\hat{r}_i \leftarrow r_j + \left\lceil \frac{D}{a(r_j+1)} \right\rceil, \tag{8}$$

where $D$ is the diameter of the graph $\mathcal{G}$. The elimination arm set $\mathcal{B}_j^r$ is constructed as

$$\mathcal{B}_j^r = \{i, i \in \mathcal{S}_j^r : \exists i' \in \mathcal{S}_j^r \setminus \{i\} \text{ such that } \tilde{\mu}_{i,j}^r \leq \tilde{\mu}_{i',j}^r - 2U_{i,j}^r\}. \tag{9}$$

The candidate set $\mathcal{S}_j$ is updated based on the information in $\mathcal{B}_j$ until only one arm remains, which is identified as the optimal arm. Once an arm is eliminated from $\mathcal{S}_j$, it is also removed from $\mathcal{B}_j$ to avoid redundant communication (Line 19, Algorithm 1).

## 4 Analysis

In this section, we summarize the theoretical results concerning both regret and communication cost. We begin by introducing Lemma 1, which provides a concentration bound for the global mean. Unlike traditional concentration results based on real-time data, our approach leverages batched information to perform consensus estimation, which obtains a similar performance. This key insight allows us to establish upper bounds on both communication rounds and regret, which are detailed in Section 4.1. Finally, Section 4.2 presents the corresponding lower bounds.

**Lemma 1.** *Let $i \in \mathcal{K}$ be any arm, $j \in \mathcal{N}$ be any agent, and $r \in \{1, 2, \dots\}$ be any epoch. Assume reward $X_{i,j}(t)$ is an i.i.d. process with unknown mean $\mu_{i,j}$. Set $\sigma_i(r_j) = \frac{2}{r_j+2}$. Then, with probability at least $1 - 2\delta$, the gap between the global estimate $\tilde{\mu}_{i,j}^r$ and the global mean $\mu_i$ satisfies*

$$\left| \bar{\mu}_{i,j}^r - \mu_i \right| \leq \sqrt{\frac{\log \delta^{-1}}{aN(r_j+1)(r_j+2)}} + \frac{2C}{(1-\lambda_2)(r_j+2)} + \frac{1}{r_j+2}, \tag{10}$$

*where $\lambda_2$ is the second largest eigenvalue of the weight matrix $\mathbf{W}$ (characterizing network connectivity), $\sigma_i(r_j)$ is the variance proxy of local means across agents, and $C$ is a positive constant determined by $\mathcal{G}$, i.e., $C = 1$ when $\mathcal{G}$ is balanced; otherwise, $C = \sqrt{N}$.*

**Proof Sketch of Lemma 1.** We decompose the estimation error into three parts: quantization error $|\bar{\mu}_{i,j}^r - \tilde{\mu}_{i,j}|$, consensus error $|\tilde{\mu}_{i,j}^r - \hat{\mu}_i^r|$, and concentration error $|\hat{\mu}_i^r - \mu_i|$. The quantization error arises from limited communication precision and is controlled by the quantization scheme. The consensus error results from averaging over a network with imperfect information propagation and is bounded using spectral properties of the communication matrix. Finally, the concentration error comes from estimating the true mean using sub-Gaussian samples and is handled via standard probabilistic bounds. Combining these yields a confidence interval for the global estimate.

### 4.1 Upper Bound

**Theorem 1** (Upper regret bound). *Let $U_{i,j}^r$ in (6) with $\delta = T^{-2}$ be the radius of the confidence interval of a random $[0,1]$-valued i.i.d. process. Given any constant $\gamma > 0$ and $\sigma_i(r_j) = 2/(r_i+2)$, Algorithm 1 could achieve the following results,*

*(i) For each agent $j$, individual regret:*

$$\mathbb{E}[R_j(T)] \leq O\left( \sum_{i:\Delta_i>0} N^{-1}\Delta_i^{-1}\log T \right),$$

*(ii) Group regret:*

$$\mathbb{E}[R(T)] \leq O\left( \sum_{i:\Delta_i>0} \Delta_i^{-1}\log T \right),$$

**Proof Sketch of Theorem 1.** To bound both types of regret, we first establish an upper bound on the sample counts of sub-optimal arms. From equation (7), we derive an instance-dependent bound, then account for the additional samples required to synchronize agents under the communication delay. This adjustment depends on the graph's diameter and the theoretical result in equation (7). Finally, combining the gap terms with these sample counts yields the overall regret bounds. Detailed proofs are provided in Appendix D.2.

**Theorem 2** (Upper communication cost). *When all agents execute `EpoInc-SE` (Algorithm 1), the total communication cost over the entire time horizon is at most $O(\sum_{i:\Delta_i>0} \sqrt{N}\Delta_i^{-1}\sqrt{\log T})$ bits, in order to achieve the regret bounds stated in Theorem 1.*

**Proof Sketch of Theorem 2.** To optimize communication cost, the key idea is to minimize both the size of global estimates and the number of communication rounds. The message size is reduced to $O(1)$ through differential communication, while the epoch-based exploration policy decreases the communication frequency to the order of $O(\sqrt{\log T})$. By combining these two strategies, we derive the upper bound of the overall communication cost.

**Remark 2.** *The lower bound of the communication cost is still an important problem for heterogeneous bandit problems. The previous work Wang et al. [2019] provides a constant level lower bound for the homogeneous bandit. However, the heterogeneous bandit setting poses a more challenging property, i.e., heterogeneous feedback, which implies that more essential communication is needed for obtaining a near-optimal result. It is an interesting work that is worth learning.*

**Remark 3.** *In the worst case, the message in the communication phase I will consume at most $O(NK)$ bits. In the communication phase II, the message will consume at most $K \log K + K \log \log T$ bits.*

## 4.2 Lower regret bound

Besides the upper bounds of regrets, we also present lower bounds for this problem. We investigate the lower bounds of both individual and group regrets, where the elements in the upper bounds match those in the lower bounds.

**Theorem 3** (Regret lower bound). *For FMAB problems with any number of agents, arms, and stochastic rewards satisfying a $1$-Gaussian distribution, if the graph $\mathcal{G}$ is connected, any federated bandit algorithm must incur regrets at least:*

*(i) For each agent $j$, individual regret:*

$$\liminf_{T\to\infty} \frac{\mathbb{E}[R_j(T)]}{\log T} \geq \sum_{i:\Delta_i>0} \frac{2}{N\Delta_i}.$$

*(ii) Group regret:*

$$\liminf_{T\to\infty} \frac{\mathbb{E}[R(T)]}{\log T} \geq \sum_{i:\Delta_i>0} \frac{2}{\Delta_i}.$$

**Proof Sketch of Theorem 3.** To derive the lower bound for federated heterogeneous bandit problems, the main challenge lies in handling the randomness of decision-making across multiple agents. To address this, we introduce an auxiliary "replica" subsystem that allows agent $j$ to access other agents' observations on the same arm. This additional information does not tighten the lower bound. We then show that it does not affect the consensus estimation, after which the regret lower bound follows directly from the proof techniques in [Lattimore and Szepesvári, 2020].

**Remark 4.** *The lower bound of the heterogeneous federated bandit problem is also discussed in Theorem 3 in Xu and Klabjan [2023]. Different from the previous work, we prove the lower bound by using a relaxed model. In this model, each agent obtains additional information (Details are given in Section 3), and the result is obtained directly by using Theorem 1 in Garivier et al. [2019].*

# 5 Experiments

In this section, we conduct a series of experiments on different algorithms. All experiments are repeated for 50 trials, with means plotted as lines and standard deviations as shaded regions.

**Setups and Baselines.** Unless otherwise stated, the experiment scenario involves a network of $N = 8$ agents and $K = 10$ arms, parameters $C = \sqrt{N}$, $a = 5$ and $T = 10^6$. To ensure a fair comparison, we use a ring graph and all agents occupy the same weight in their neighborhoods, which is a common undirected connected graph with the second largest eigenvalue $\lambda_2 = 0.5713$. We consider four baselines: `Gossip_UCB` [Zhu et al., 2021], `Dis-UCB` [Zhu and Liu, 2023], `Fed2-UCB` [Shi and Shen, 2021] and `UCB-TCOM` [Wang and Yang, 2023]. The theoretical results of them are discussed in Table 1. The individual regret is evaluated by the maximum individual regret among all agents. For `Fed2-UCB` and `UCB-TCOM`, a consensus estimation module is attached to them to help them against heterogeneity. The two algorithms communicate the historical reward at each time slot. Hence, the consensus estimator is $\hat{\mu}_i(t) = \frac{1}{Nt} \sum_{j=1}^{N} \sum_{k=1}^{t} X_{i,j}(k)$.

**Observations.** From Figures 1a and 1b, both group and individual regrets of `EpoInc-SE` are only slightly worse than that of `Fed2-UCB`, primarily because `Fed2-UCB` leverages a central server, which enables faster convergence compared to a fully distributed graph. Figure 1c shows that `EpoInc-SE` results in the lowest bit-level communication overhead among all compared methods, highlighting the efficiency of our algorithm in minimizing communication costs. Figures 1d-1f indicate that `Epoinc-SE` still has good performance when $N, K, \delta$ vary. We also make some experiments under homogeneous settings in Section E and some experiments in large-scale multi-agent systems in Section F.

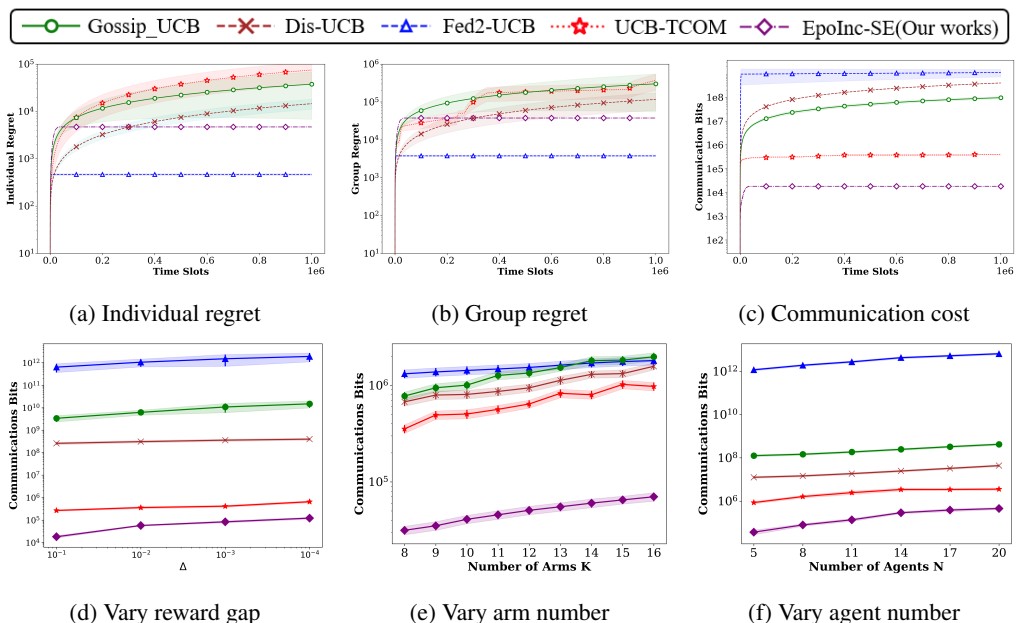

Figure 1: Performance comparison with different algorithms.

# 6 Conclusion

This paper proposed a communication-saving solution to address the challenges faced by FMAB problems in environments with heterogeneous rewards and fully distributed communication networks. We successfully tackled the shortcomings of existing research in areas such as information fusion, regret performance, and communication efficiency, particularly in scenarios where there is no central coordinator and agents can only communicate with their neighbors. The main contribution of this paper lies in the design and introduction of an efficient federated learning algorithm `EpoInc-SE`, which greatly reduces the communication cost and obtains $O(1)$ bits. Besides, we first propose a lower regret bound in this setting and prove that `EpoInc-SE` is near-optimal under limited communication. Future work could explore applying this framework to more complex network dynamics or considering more advanced privacy-preserving mechanisms.

# 7 Acknowledgment

This work was supported by NSFC (no. 62306138), JiangsuNSF (no. BK20230784), and the Innovation Program of State Key Laboratory for Novel Software Technology at Nanjing University (no. ZZKT2025B25). HQ acknowledges the financial support from the EU Horizon CL4-2022-HUMAN-02 research and innovation action under grant agreement 101120237, project ELIAS (European Lighthouse of AI for Sustainability) and from the One Health Action Hub, University Task Force for the resilience of territorial ecosystems, funded by Università degli Studi di Milano (PSR 2021-GSA-Linea 6).

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

# A   Appendix / Symbol explanation

For all symbols in this article, we give explanations of them in Table 3.

| Symbol/Term | Definition |
|---|---|
| $\mathcal{G}(\mathcal{N}, \mathcal{E}, \mathcal{A})$ | A graph to describe a multi-agent system |
| $\mathcal{N} = \{1, \ldots, N\}$ | Set of agents in a multi-agent system |
| $\mathcal{E} \subset \mathcal{N} \times \mathcal{N}$ | The edge set in graph $\mathcal{G}$ |
| $\mathcal{A} = [a_{i,j}]_{N \times N}$ | The weight matrix to describe the relations between agents |
| $\mathcal{N}_j$ | Neighborhood of agent $j$, excluding agent $j$ |
| $\boldsymbol{W} = [\omega_{a,b}]_{N \times N}$ | Communication matrix |
| $D$ | The diameter of graph $\mathcal{G}$ |
| $\lambda_2$ | The second largest eigenvalue of $\boldsymbol{W}$ |
| $C$ | A symbol to describe whether the graph is balance |
| $\mathcal{K} = \{1, \ldots, K\}$ | Set of arms in a multi-armed bandit (MAB) problem |
| $T$ | Total number of time slots |
| $A_j(t)$ | Arm chosen by agent $j$ at time slot $t$ |
| $X_{A_j(t),j}(t)$ | Random reward received by agent $j$ after pulling arm $A_j(t)$ at time slot $t$ |
| $X_{i,j}(t)$ | Random reward for arm $i$ observed by agent $j$ at time slot $t$ |
| $\mu_{i,j}$ | Mean reward for arm $i$ observed by agent $j$, bounded in $[0, 1]$ |
| $X_i(t)$ | Global reward for arm $i$ at time slot $t$ |
| $a$ | Constant that determines the size of an epoch |
| $p^r$ | The sample count of any arm in $\mathcal{S}_j^r$ in epoch $r_j$, $p^r = a(r_j + 1)$ |
| $\overline{X}_{i,j}^r$ | The average local reward of $X_{i,j}$ in epoch $r_j$ |
| $\mu_i$ | Global mean reward for arm $i$ |
| $\tilde{\mu}_{i,j}(t)$ | The global estimate of agent $j$ on arm $i$ at time slot $t$ |
| $i^\star$ | The unique optimal arm with the largest global mean reward |
| $i^{\mathrm{max}}$ | $i^{\mathrm{max}} = \arg\max_{i \in \mathcal{S}_j} \tilde{\mu}_{i,j}$ |
| $\mathcal{S}_j^r$ | The candidate arm set of agent $j$ at time slot $t$ |
| $\mathcal{B}_j^r$ | The elimination arm set of agent $j$ at time slot $t$ |
| $\hat{r}_i$ | The epoch label attached to arm $i$ |
| $\Delta_i = \mu_{i^\star} - \mu_i$ | Reward gap between the optimal arm and arm $i$ |
| $\tau_{i,j}^r$ | The sample count of agent $j$ on arm $i$ until epoch $r_j$ |
| $\tau_i^r$ | The global sample count on arm $i$ until epoch $r_j$ |
| $\delta$ | The violation probability of confidence interval |
| $U_{i,j}^r(\delta)$ | The radius of confidence interval |
| $\mathbb{E}[R(T)]$ | Expected group regret |
| $\mathbb{E}[R_j^T(\mathcal{A})]$ | Expected individual regret of agent $j$ |

Table 3: Summary of symbols and Definitions

# B   Appendix / some knowledge of graphs

Throughout this paper, we consider a FMAB problem with $N$ agents operating in a time-invariant network. The network is represented by a communication graph $\mathcal{G}(\mathcal{N}, \mathcal{E}, \mathcal{A})$, which consists of three components:

1. $\mathcal{N} = \{1, \ldots, N\}$ is the set of agents in the network, corresponding to the number of agents in the distributed system.

2. $\mathcal{E} \subset \mathcal{N} \times \mathcal{N}$ is the edge set, which determines the connectivity between agents.

3. $\mathcal{A} = [a_{j,j'}]_{N \times N}$ is the adjacency matrix of the graph $\mathcal{G}$, where $a_{j,j'}$ denotes the weight of the edge between agents $j$ and $j'$.

Notably, the adjacency matrix represents the importance of one agent to its neighbors and encodes neighborhood information in $\mathcal{G}$. Specifically, $a_{j,j'}$ is the weight from agent $j'$ to agent $j$. Since the graph is directed, we have $a_{j,j'} \neq a_{j',j}$.

The graph has no self-loops, meaning that $a_{j,j} = 0$ for all $j \in \mathcal{N}$. An edge between agents $j$ and $j'$ exists if and only if $a_{j,j'} \neq 0$, i.e., $(j, j') \in \mathcal{E}$.

For each agent $j$, its neighborhood is denoted as $\mathcal{N}_j = \{j' \mid j' \in \mathcal{N}, a_{j,j'} \neq 0, j' \neq j\}$. Finally, we define the diameter of the graph $\mathcal{G}$ as $D$, which represents the longest distance between any two agents in the network.

For graph $\mathcal{G}$, its corresponding Laplacian matrix $\mathcal{L}$ is defined as follows

$$
\mathcal{L} = \begin{cases} -a_{j,j'}, & j \neq j' \\ \sum_{j'=1}^{N} a_{j,j'}. & j = j' \end{cases}
$$

The maximum degree of graph $\mathcal{G}$ is defined as $\epsilon = \max_i(\sum_{j' \neq j} a_{j,j'})$. Then, for any constant $\beta \in (0, 1/\epsilon]$, the Perron matrix $W = I - \beta\mathcal{L}$ could be obtained. The Perron matrix $\boldsymbol{W} = [\omega_{i,j}]_{N \times N}$ is a doubly random matrix and both the sum of row elements and column elements in $\boldsymbol{W}$ is 1. In the multi-agent bandit setting, it is widely used to solve the consensus problem Olfati-Saber et al. [2007].

## C   Appendix / Preliminaries of the problem

**Lemma 2.** *[Yan et al., 2012] For a doubly random matrix $\boldsymbol{W}$, it is an irreducible, doubly stochastic matrix with strictly positive diagonal entries, then there exists a positive constant $C$ such that*

$$
\sum_{j=1}^{N} |\omega_{i,j}^k - \frac{1}{N}| < C\lambda_2^k, \tag{11}
$$

*where $\omega_{i,j}^k$ represents the element in the $i$-th row and $j$-th column of the matrix $W^k$, $k$ represents the iteration step and $\lambda_2$ is the second largest value of matrix $\boldsymbol{W}$. $C$ is equal to 1 if the graph $\mathcal{G}$ is balanced, otherwise, $C = \sqrt{N}$.*

**Lemma 3.** *[Molloy and Reed, 2002] Assume that $X(t) - \mu$ is independent, $\sigma$ sub-Gaussian random variable. Then for any $\epsilon \geq 0$,*

$$
\mathbb{P}(\hat{\mu} \geq \mu + \epsilon) \leq \exp(-\frac{n\epsilon^2}{2\sigma^2}),
$$

$$
\mathbb{P}(\hat{\mu} \leq \mu - \epsilon) \geq \exp(-\frac{n\epsilon^2}{2\sigma^2}),
$$

*where $\hat{\mu} = \frac{1}{n}\sum_{t=1}^{n} X(t)$ and $n$ is the sample count.*

**Lemma 4.** *[Lattimore and Szepesvári, 2020] Suppose that $X_i$ is $\sigma_i^2$ sub-Gaussian and $X_i$ are all independent for $i \in \{1, \ldots, N\}$. Then we have $\frac{1}{N}\sum_{i=1}^{N} X_i$ is $\frac{\sum_i^N \sigma_i^2}{N^2}$ sub-Gaussian.*

**Lemma 5.** *[Dubhashi and Panconesi, 2009] If a random variable $X$ has a finite mean and $a \leq X \leq b$ almost surely, then $X$ is $\frac{1}{4}(b-a)^2$ sub-Gaussian.*

## D   Appendix / Missed proofs

### D.1   Proof of Lemma 1

*Proof.* The goal is to obtain an unbiased estimation $\bar{\mu}_{i,j}^r$ of the global mean $\mu_i$. To achieve the goal, we could divide the problem into three parts: $|\bar{\mu}_{i,j}^r - \tilde{\mu}_{i,j}|$, $|\tilde{\mu}_{i,j}^r - \hat{\mu}_i|$ and $|\hat{\mu}_i - \mu_i|$. According to the triangle inequality, we have

$$
|\bar{\mu}_{i,j}^r - \mu_i| \leq |\bar{\mu}_{i,j}^r - \tilde{\mu}_{i,j}| + |\tilde{\mu}_{i,j}^r - \hat{\mu}_i^r| + |\hat{\mu}_i^r - \mu_i|. \tag{12}
$$

The first term $|\bar{\mu}_{i,j}^r - \tilde{\mu}_{i,j}|$ is bounded according to the definition of information quantization introduced in Section 3.3. We have

$$
|\bar{\mu}_{i,j}^r - \tilde{\mu}_{i,j}^r| \leq \frac{1}{r_j + 2}.
$$

Since all agents make decisions synchronously, we remove the subscript $j$ from the epoch number $r_j$. The consensus process mainly acts at the end of each epoch; therefore, we consider the average reward $\bar{X}_{i,j}^r$ during the total epoch instead of the real-time reward $X_{i,j}(t)$ in each time slot $t$. To facilitate matrix-based operations, we stack equation (5) as follows

$$\tilde{\boldsymbol{\mu}}_{\boldsymbol{i}}^{\boldsymbol{r}} = (1 - \sigma_i(r))\boldsymbol{W}\tilde{\boldsymbol{\mu}}_{\boldsymbol{i}}^{\boldsymbol{r-1}} + \sigma_i(r)\bar{\boldsymbol{X}}_{\boldsymbol{i}}^{\boldsymbol{r}}, \qquad (13)$$

where $\tilde{\boldsymbol{\mu}}_{\boldsymbol{i}}^{\boldsymbol{r}}$ and $\bar{\boldsymbol{X}}_{\boldsymbol{i}}^{\boldsymbol{r}}$ are defined as

$$\tilde{\boldsymbol{\mu}}_{\boldsymbol{i}}^{\boldsymbol{r}} = [\tilde{\mu}_{i,1}^r, \ldots, \tilde{\mu}_{i,N}^r]^T,$$
$$\bar{\boldsymbol{X}}_{\boldsymbol{i}}^{\boldsymbol{r}} = [\bar{X}_{i,1}^r, \ldots, \bar{X}_{i,N}^r]^T.$$

The coefficient $\sigma_i(r_j)$ represents the ratio of the total amount of data aggregated by variable $\bar{X}_{i,j}^r$ to the overall data volume. The iteration of (5) occurs at the end of each epoch $r_j$, which implies that the sample count could be connected to epoch $r$. Hence, we define $\sigma_i(r_j) = \frac{2a(r_j+1)}{a(r_j+1)(r_j+2)} = \frac{2}{r_j+2}$, where $a$ is the pre-trained sample count at the beginning of the algorithm.

According to the state above, iterating (13) at each epoch $r$ yields that

$$\begin{aligned}
\tilde{\boldsymbol{\mu}}_{\boldsymbol{i}}^{\boldsymbol{r}} &= \frac{r}{r+2}\boldsymbol{W}\tilde{\boldsymbol{\mu}}_{\boldsymbol{i}}^{\boldsymbol{r-1}} + \frac{2}{r+2}\bar{\boldsymbol{X}}_{\boldsymbol{i}}^{\boldsymbol{r}} \\
&= \frac{2}{a(r+1)(r+2)}\boldsymbol{W}^r\tilde{\boldsymbol{\mu}}_{\boldsymbol{i}}^{\boldsymbol{0}} + \frac{2}{a(r+2)(r+1)}\sum_{k=1}^{r}\boldsymbol{W}^{r-k}a(k+1)\bar{\boldsymbol{X}}_{\boldsymbol{i}}^{\boldsymbol{k}} \qquad (14) \\
&\overset{(a)}{=} \frac{2}{a(r+2)(r+1)}\sum_{k=0}^{r}\boldsymbol{W}^{r-k}a(k+1)\bar{\boldsymbol{X}}_{\boldsymbol{i}}^{\boldsymbol{k}},
\end{aligned}$$

where equation (a) is obtained by setting $\tilde{\boldsymbol{\mu}}_{\boldsymbol{i}}^{\boldsymbol{0}} = \bar{\boldsymbol{X}}_{\boldsymbol{i}}^{\boldsymbol{0}}$.

Then, separate agent $j$'s global estimate from the vector $\tilde{\boldsymbol{\mu}}_{\boldsymbol{i}}^{\boldsymbol{r}}$, we have

$$\tilde{\mu}_{i,j}^r = \frac{2}{a(r+2)(r+1)}\sum_{k=0}^{r}\sum_{j'=1}^{N}\omega_{j,j'}^{r-k}a(k+1)\bar{X}_{i,j'}^k. \qquad (15)$$

For the fully connected graph, each agent can observe all other agents' observations. Hence, their global estimates $\hat{\mu}_i^r$ on arm $i$ are the same, which is written as

$$\begin{aligned}
\hat{\mu}_i^r &= \frac{2}{a(r+2)(r+1)}\sum_{\tau=0}^{t}\frac{1}{N}\sum_{j=1}^{N}X_{i,j}(\tau) \\
&\overset{(a)}{=} \frac{2}{a(r+2)(r+1)}\sum_{k=0}^{r}\frac{1}{N}\sum_{j=1}^{N}a(k+1)\bar{X}_{i,j}^k,
\end{aligned} \qquad (16)$$

where equation (a) is obtained because the concentration accrues at the end of each epoch, i.e., $t = a(r+1)(r+2)/2$.

The error between $\tilde{\mu}_{i,j}^r$ and $\hat{\mu}_i^r$ is

$$\begin{aligned}
|\tilde{\mu}_{i,j}^r - \hat{\mu}_i^r| &= \frac{2}{a(r+1)(r+2)}\sum_{k=0}^{r}\sum_{j'=1}^{N}(\omega_{j,j'}^{r-k} - \frac{1}{N})a(k+1)\bar{X}_{i,j'}^k \\
&\overset{(a)}{\leq} \frac{2}{(r+1)(r+2)}\sum_{k=0}^{r}C\lambda_2^{r-k}(1+k) \\
&\overset{(b)}{\leq} \frac{2C}{(r+1)(r+2)} \times \frac{\lambda_2^{r+2} - \lambda_2}{(1-\lambda_2)^2} + \frac{2C}{(r+2)(1-\lambda_2)} \\
&\leq \frac{2C}{(1-\lambda_2)(r+2)},
\end{aligned} \qquad (17)$$

where equation (a) yields from Lemma 2 and equation (b) is derived based on the properties of arithmetic and geometric sequences.

For a federated setting with $N$ agents, the global reward is defined as $X_i(t) = \frac{1}{N} \sum_{j=1}^{N} X_{i,j}(t)$, where $X_{i,j}(t) \in [0, 1]$ represents the local reward observed by agent $j$. According to Lemma 5, each $X_{i,j}(t)$ is $\frac{1}{4}$-sub-Gaussian. By applying Lemma 4, which addresses linear combinations of sub-Gaussian variables, it follows that the global reward $X_i(t)$ is $\frac{1}{4N}$-sub-Gaussian.

By using Lemma 3 and setting $\varepsilon = \sqrt{\frac{\log \delta^{-1}}{2N\tau}}$ and $\sigma^2 = \frac{1}{4N}$, the probability that the estimate $\hat{\mu}_i^r$ exceeds the confidence interval after round $r$ is given by

$$
\mathbb{P}\left( \hat{\mu}_i(t) \geq \mu_i + \sqrt{\frac{\log \delta^{-1}}{aN(r+1)(r+2)}} \right) \leq \delta,
$$
$$
\mathbb{P}\left( \hat{\mu}_i(t) \leq \mu_i - \sqrt{\frac{\log \delta^{-1}}{aN(r+1)(r+2)}} \right) \leq \delta,
$$
(18)

i.e., with probability at least $1 - 2\delta$, we have

$$
|\hat{\mu}_i^\tau - \mu_i| \leq \sqrt{\frac{\log \delta^{-1}}{aN(r+1)(r+2)}}.
$$
(19)

Combining the upper bounds of the three terms, we obtain the following result based on equation (12):

$$
|\bar{\mu}_{i,j}^r - \mu_i| \leq \sqrt{\frac{\log \delta^{-1}}{aN(r+1)(r+2)}} + \frac{2C}{(1-\lambda_2)(r+2)} + \frac{1}{r+2}.
$$

which is the radius of the confidence interval. □

### D.2 Proof of Theorem 1

*Proof.* To establish bounds on both regret and communication cost, the key step is to characterize the sample complexity of all agents. Hence, the proof is divided into three parts: (1) bounding the total number of samples collected by each agent, (2) deriving the corresponding regret bounds, and (3) analyzing the total communication cost.

**Bound the sample counts:** Since Algorithm 1 is embedded in each agent and executed in a similar manner, we focus our analysis on a representative agent $j$. Agent $j$ usually distinguishes whether an arm is suboptimal at the end of each epoch. Recall that for all arms $i \in \mathcal{K} \setminus \{i^\star\}$, $\Delta_i > 0$. Agents learn the gap $\Delta_i > 0$ and continuously eliminate the arms $i : \Delta_i > 0$. To accelerate the update of the candidate set $\mathcal{S}_j$, detection is performed at the end of each epoch. Once suboptimal arm $i$ is identified through this detection, agent $j$ will add it into the elimination arm set $\mathcal{B}_j^r$ and later initiate a communication phase that broadcasts the elimination set $\mathcal{B}_j$ across the distributed system in waves.

Since that $\tilde{\mu}_{i,j}^r$ is to estimate $\mu_i$, the reward gap $\Delta_i$ for each agent $j \in \mathcal{N}$ is related to $U_{i,j}^r$. The gap between $\tilde{\mu}_{i^{\max},j}^r$ and $\tilde{\mu}_{i,j}^r$ is

$$
2U_{i,j}^r \overset{(7)}{\geq} \tilde{\mu}_{i^{\max},j}^r - \tilde{\mu}_{i,j}^r \overset{(a)}{\geq} \Delta_i - 2U_{i,j}^r
$$
(20)

where inequality (a) is because $\tilde{\mu}_{i^{\max},j}^r \geq \tilde{\mu}_{i^\star,j}^r \geq \mu_{i^\star} - U_{i,j}^r$ and $\tilde{\mu}_{i,j}^r \leq \mu_i + U_{i,j}^r$.

Denote $A_{i,j,t}$ as the event of agent $j$ pulling arm $i$ at the time slot $t$, then we have

$$
\mathbb{P}\left( \bigcap_{i,j,t} A_{i,j,t} \right) = 1 - \mathbb{P}\left( \bigcup_{i,j,t} \neg A_{i,j,t} \right) \geq 1 - \sum_{i,j,t} \mathbb{P}\left( \neg A_{i,j,t} \right) \geq 1 - 2tNK\delta.
$$

At the end of each epoch $r_j$, we take a detection for each arm in the candidate set $\mathcal{S}_j^r$. The corresponding sample count is equal to $\tau_{i,j} = a(r_j + 1)(r_j + 2)/2$. Equation (20) is written as

$$
\Delta_i \leq 4U_{i,j}^r \leq 4\sqrt{\frac{\log \delta^{-1}}{aN(r+1)(r+2)}} + \frac{\frac{8C}{1-\lambda_2} + 4}{r_j + 2},
$$

i.e.,

$$\Delta_i^2 - \frac{8\Delta_i(\frac{2C}{1-\lambda_2}+1)}{r+2} + \frac{16(\frac{2C}{1-\lambda_2}+1)^2}{(r+2)^2} \le \frac{16\log\delta^{-1}}{aN(r+1)(r+2)},$$

$$(r+1)(r+2-\frac{8(\frac{2C}{1-\lambda_2}+1)}{\Delta_i}) \le \frac{16\log\delta^{-1}}{aN\Delta_i^2},$$

$$r \le \frac{4\sqrt{\log\delta^{-1}}}{\sqrt{aN}\Delta_i} - 2 + \frac{8(\frac{2C}{1-\lambda_2}+1)}{\Delta_i}.$$

According to the definition of the epoch label $\hat{r}_i$ in equation (8), it follows that any suboptimal arm $i$ will be added into the elimination set $\mathcal{B}_j^r$ no later than $\left\lceil \frac{4\sqrt{\log\delta^{-1}}}{\sqrt{aN}\Delta_i} - 2 + \frac{8(\frac{2C}{1-\lambda_2}+1)}{\Delta_i} \right\rceil$. In the next epoch $\left\lceil \frac{4\sqrt{\log\delta^{-1}}}{\sqrt{aN}\Delta_i} - 2 + \frac{8(\frac{2C}{1-\lambda_2}+1)}{\Delta_i} \right\rceil + 1$, agent $j$ will broadcast arm $i$ to all neighbors (Communication phase I).

As the number of time slots per epoch grows over time, the number of epochs needed for information to propagate from a single agent to the entire network decreases accordingly. The maximum communication epoch required is determined by

$$\min_{\tilde{r}} \sum_{r=1}^{\tilde{r}} a(r+1) \ge D, \tag{21}$$

where $\tilde{r}$ represents the most epoch number. From the optimization problem (21), the epoch number is at most $\left\lceil \sqrt{\frac{2D}{a}+2} \right\rceil$ to broadcast the message to all agents. In Algorithm 1, the epoch for sampling suboptimal arm $i$ is donated by $\bar{r}_i = \left\lceil \frac{4\sqrt{\log\delta^{-1}}}{\sqrt{aN}\Delta_i} - 2 + \frac{8(\frac{2C}{1-\lambda_2}+1)}{\Delta_i} \right\rceil + \left\lceil \sqrt{\frac{2D}{a}+2} \right\rceil$.

**Bound the regrets:** According to the basic regret decomposition identity, the individual regret in equation (1) of agent $j$ could be written as

$$R_j(T) = T\mu_{i^\star} - \sum_{t=1}^{T} \mathbb{E}[X_{A_j(t)}(t)] = T\mu_{i^\star} - \sum_{t=1}^{T} \mu_{A_j(t)}$$

$$\le \sum_{i:\Delta_i>0} \Delta_i \tau_{i,j}(T) = \sum_{i:\Delta_i>0} \frac{a\Delta_i(\bar{r}_i+1)(\bar{r}_i+2)}{2}$$

$$\le \sum_{i:\Delta_i>0} \frac{a\Delta_i}{2}\left(\frac{4\sqrt{\log\delta^{-1}}}{\sqrt{aN}\Delta_i} + \bar{d}_i\right)^2$$

$$\le \sum_{i:\Delta_i>0} \left(\frac{16\log T}{N\Delta_i} + \frac{2\bar{d}_i\sqrt{2a\log T}}{\sqrt{N}} + \frac{a\bar{d}_i^2}{2}\right),$$

where

$$\bar{d}_i = \frac{16C+8-8\lambda_2}{\Delta_i(1-\lambda_2)}.$$

According to the definition of group regret and the individual regret bound, equation (2) is written as

$$R(T) = \sum_{j=1}^{N} R_j(T) \le \sum_{i:\Delta_i>0} \left(\frac{16\log T}{\Delta_i} + 2\bar{d}_i\sqrt{2aN\log T} + \frac{aN\bar{d}_i^2}{2}\right).$$

$\square$

## D.3 Proof of Theorem 2

*Proof.* The communication in Algorithm 1 has two types, i.e., Communication phase I and II. In Communication phase I, agents mainly share the indexes of sub-optimal arms and their corresponding

epoch label $\hat{r}_i$. In Communication phase II, agents mainly share differential information $\delta_{i,j}^r$ about the global estimates.

**Consider the size of each message:** There are three types of information that need to be communicated in this setting: (a) the arm set $\mathcal{B}_j^r$, (b) the epoch label $\hat{r}_i$, and (c) the global estimate $\delta_{i,j}^r$. The message size for type (a) requires only $O(\log K)$ bits. For type (b), the size depends on the epoch label $\hat{r}_i$, which is upper bounded by $\bar{r}_i$ and can be encoded using $O(\sqrt{\log T})$ bits, as shown in the proof of Theorem 1. For type (c), the global estimate $\delta_{i,j}^r$ lies in $[0, 1]$ and is adaptively quantized. It is the most important part that needs to be optimized, as it dominates the total communication cost. The key challenge is to transmit the order of the global estimate from $O(\log(T))$ to $O(1)$.

After agent $j$ computes the global estimate $\tilde{\mu}_{i,j}^r$ at the end of epoch $r_j$, it will be transmitted into a more concise version $\bar{\mu}_{i,j}^r$ with $\lceil 1 + \log_2 r_j \rceil$ bits. Then, the difference is calculated from $\delta_{i,j}^r = \bar{\mu}_{i,j}^r - \bar{\mu}_{i,j}^{r-1}$. Hence, we could bound $\delta_{i,j}^r$ to reduce the total communication cost. The difference $\delta_{i,j}^r$ holds that

$$
\begin{aligned}
|\delta_{i,j}^r| = |\bar{\mu}_{i,j}^r - \bar{\mu}_{i,j}^{r-1}| &= |\bar{\mu}_{i,j}^r - \tilde{\mu}_{i,j}^r - \bar{\mu}_{i,j}^{r-1} + \tilde{\mu}_{i,j}^{r-1} + \tilde{\mu}_{i,j}^r - \tilde{\mu}_{i,j}^{r-1}| \\
&\leq |\bar{\mu}_{i,j}^r - \tilde{\mu}_{i,j}^r| + |\bar{\mu}_{i,j}^{r-1} - \tilde{\mu}_{i,j}^{r-1}| + |\tilde{\mu}_{i,j}^r - \tilde{\mu}_{i,j}^{r-1}| \overset{(a)}{\leq} \frac{1}{r} + \frac{1}{r-1} + |\tilde{\mu}_{i,j}^r - \tilde{\mu}_{i,j}^{r-1}|,
\end{aligned}
\tag{22}
$$

where equation $(a)$ yields from the quantization process in Algorithm 2. Due to that $\bar{\mu}_{i,j}^r$ contains $\lceil 1 + \log_2 r_j \rceil$ bits, which implies that there is a quantization error at most $1/r_j$ compared with the true estimate value $\tilde{\mu}_{i,j}^r$. For the term $|\tilde{\mu}_{i,j}^r - \tilde{\mu}_{i,j}^{r-1}|$, we have

$$
\begin{aligned}
|\tilde{\mu}_{i,j}^r - \tilde{\mu}_{i,j}^{r-1}| = |\tilde{\mu}_{i,j}^r - \hat{\mu}_i^r - \tilde{\mu}_{i,j}^{r-1} + \hat{\mu}_i^{r-1} + \hat{\mu}_i^r - \hat{\mu}_i^{r-1}| \\
\leq |\tilde{\mu}_{i,j}^r - \hat{\mu}_i^r| + |\tilde{\mu}_{i,j}^{r-1} - \hat{\mu}_i^{r-1}| + |\hat{\mu}_i^r - \hat{\mu}_i^{r-1}| \\
\leq \frac{2C}{(1 - \lambda_2)(r_j + 2)} + \frac{2C}{(1 - \lambda_2)(r_j + 1)} + |\hat{\mu}_i^r - \hat{\mu}_i^{r-1}|,
\end{aligned}
$$

where

$$
\hat{\mu}_i^r - \hat{\mu}_i^{r-1} = \frac{\sum_{t=0}^{\frac{a(r_j+1)(r_j+2)}{2}} X_i(t)}{a(r_j+1)(r_j+2)/2} - \frac{\sum_{t=0}^{\frac{ar_j(r_j+1)}{2}} X_i(t)}{ar_j(r_j+1)/2}.
$$

Due to that $X_i(t)$ is $\frac{1}{2\sqrt{N}}$-sub-Gaussian random variable, $\hat{\mu}_i^r - \hat{\mu}_i^{r-1}$ could be considered as $\sqrt{\frac{1}{2aN(r_j+1)(r_j+2)}}$-sub-Gaussian random variable.

Thus, we can further derive that, for a dummy variable $x \geq \sqrt{\ln 2}$,

$$
\begin{aligned}
\mathbb{P}\left(|\hat{\mu}_i^r - \hat{\mu}_i^{r-1}| \geq \sqrt{\frac{x^2}{2aN(r_j+1)(r_j+2)}}\right) &\leq 2\exp\left[-\frac{\frac{a(r_j+1)(r_j+2)}{2} \times \frac{x^2}{2a(r_j+1)(r_j+2)}}{2 \times \frac{1}{4N}}\right] \\
&= 2\exp[-x^2/2],
\end{aligned}
$$

which implies that

$$
\mathbb{P}\left(|\delta_{i,j}^r| \geq \frac{1}{r_j} + \frac{1}{r_j - 1} + \frac{2C}{(1-\lambda_2)(r_j+2)} + \frac{2C}{(1-\lambda_2)(r_j+1)} + \sqrt{\frac{x^2}{2aN(r_j+2)^2}}\right) \leq 2\exp[-x^2/2]
$$

$$
\Rightarrow \mathbb{P}\left(|\delta_{i,j}^r| \geq \frac{2 + \frac{4C}{1-\lambda_2} + \frac{x}{\sqrt{2aN}}}{r_j + 2}\right) \leq 2\exp[-x^2/2]
$$

$$
\overset{(a)}{\Rightarrow} \mathbb{P}\left(L_{i,j}^r \geq 3 + \log_2 r_j + \log_2 \frac{2 + 4C/(1-\lambda_2) + x/\sqrt{2aN}}{r_j + 2}\right) \leq 2\exp[-x^2/2]
$$

$$
\Rightarrow \mathbb{P}\left(L_{i,j}^r \geq 3 + \log_2(2 + 4C/(1-\lambda_2) + x/\sqrt{2aN})\right) \leq 2\exp[-x^2/2]
$$

$$
\overset{(b)}{\Rightarrow} \mathbb{P}\left(L_{i,j}^r \leq 3 + \log_2(2 + 4C/(1-\lambda_2) + x/\sqrt{2aN})\right) \geq 1 - 2\exp[-x^2/2],
$$

where $L_{i,j}^r$ in implication (a) represents the length of the truncated version $|\delta_{i,j}^r|$ and it is bounded by

$$L_{i,j}^r + \lfloor \log_2(1/|\delta_{i,j}^r|) \rfloor \le \lceil 1 + \log_2 r_j \rceil. \tag{23}$$

Denote that $l = 3 + \log_2(2 + 4C/(1 - \lambda_2) + x/\sqrt{2aN})$, i.e., $x = \sqrt{2aN}(2^{l-3} - 2 - \frac{4C}{1-\lambda_2})$, and (b) could be written as

$$\mathbb{P}\left(L_{i,j}^r \le l\right) \ge 1 - 2\exp[-x^2/2] \ge 1 - 2\exp\left[-aN(2^{l-3} - 2 - \frac{4C}{1-\lambda_2})^2\right]. \tag{24}$$

With the above results and viewing $L_{i,j}^r$ as a random variable, we have that its cumulative distribution function (CDF) $F_{L_{i,j}^r}(l)$ satisfies the following property:

$$\forall l \ge \hat{l} = \left\lceil 3 + \log_2(2 + 4C/(1 - \lambda_2) + \sqrt{\frac{\ln 2}{2aN}}\right\rceil,$$

$$F_{L_{i,j}^r}(l) = 1 - 2\exp\left[-aN\left(2^{l-3} - 2 - \frac{4C}{1-\lambda_2}\right)^2\right].$$

Using the property of CDF, we can bound the expectation of $L_{i,j}^r$ as

$$\begin{aligned}
\mathbb{E}[L_{i,j}^r] &= \sum_{l=0}^{\infty}(1 - F_{L_{i,j}^r}(l)) \\
&\le 1 + \hat{l} + \sum_{l=\hat{l}}^{\infty}(1 - F_{L_{i,j}^r}(l)) \\
&\le 1 + \hat{l} + \int_{l=\hat{l}}^{\infty} 2\exp\left[-aN\left(2^{l-3} - 2 - \frac{4C}{1-\lambda_2}\right)^2\right] dl \\
&\le 2 + \hat{l}.
\end{aligned} \tag{25}$$

Thus, we have that in expectation, the truncated version of $|\delta_{i,j}^r|$ has a length that is less than $2 + \hat{l}$ bits.

**Consider the total number of communication rounds in communication phase I.** Communication phase I is a novel phase, used to broadcast the indices of sub-optimal arms. When an arm is identified as sub-optimal, the arm index will be exchanged at most $\left\lceil \frac{D}{a(r_j+1)} \right\rceil a(r_j + 1)$ rounds. The total communication rounds on the arm $i$ is at most $(K-1)(N-1)^2 \left\lceil \frac{D}{a(r_j+1)} \right\rceil a(r_j + 1)$ because agent $j_1$ may share the information with $j_2$ even if the information is from $j_2$ initially. Considering $N$ agents pulling $K$ arms, the number broadcast round during the total horizon $T$ in phase I is at most $(K-1)(N-1)^2 \left\lceil \frac{D}{a(r_j+1)} \right\rceil a(r_j + 1)$.

**Consider the total number of communication rounds in communication phase II.** According to Theorem 1, each suboptimal arm is pulled for at most $\bar{r}_i$ epochs. In the structure of Algorithm 1, agents exchange their differential information $\delta_{i,j}^r$ at the end of each epoch. Therefore, the total number of communication rounds in Communication Phase II is upper bounded by $N\sum_{i=1}^{K} \bar{r}_i$.

According to the discussion above, the total communication cost of the entire system is bounded by

$$\begin{aligned}
C(T) &\le (2 + \hat{l})N\sum_{i=1}^{K}\bar{r}_i + (\log_2 \hat{r}_i + \log_2 K)(K-1)(N-1)^2 \left\lceil \frac{D}{a(r_j + 1)} \right\rceil a(r_j + 1) \\
&\le O(\sum_{i:\Delta_i>0} \sqrt{N}\Delta_i^{-1}\sqrt{\log T}).
\end{aligned} \tag{26}$$

$\square$

### D.4 Proof of Theorem 3

*Proof.* We will present the proof of the lower bound in three parts.

**Problem setting.** First of all, we need to consider a more relaxed scenario. Consider a multi-agent system with $N$ agents, which is a fully connected graph. For each agent $j$, suppose that there is an associated "replica" subsystem $\hat{j}$ that can provide samples of agents $\{1, \ldots, N\} \setminus \{j\}$. When agent $j$ samples arm $i$ and obtains a reward $X_{i,j}$, replica subsystem $\hat{j}$ provides rewards $X_{i,k}, k \in \{1, \ldots, N\} \setminus \{j\}$. This construction allows $j$ to leverage peer observations without requiring global synchronization. In this construction, all agents in $\mathcal{N}$ could make different decisions, which means that the lower bound is compatible with all bandit algorithms.

**Consensus estimation evaluation.** Then, we need to ensure the convergence of consensus estimation in this setting. In each round, the agents obtain not only the global estimates from their neighbors but also information from their replica subsystems. From the replica subsystem, agent $j$ could obtain a global reward and denote it by $\bar{X}_{i,j}^\tau$. The concentration function for the fully connected graph is

$$\hat{\mu}_i^\tau = \frac{1}{\tau+1} \sum_{j=1}^{N} \sum_{k=0}^{\tau} \frac{1}{N} \bar{X}_{i,j}^k.$$

Considering that $X_{i,j}$ are all 1-sub-Gaussian variables, the linear combination reward $\hat{\mu}_i$ is a $\sqrt{\frac{\sum_{j=1}^{N} \sum_{i=1}^{N} \sigma^2}{N^4}} = \frac{1}{N}$-sub-Gaussian variable. By using Lemma 3, the relationship between $\hat{\mu}_i^\tau$ and $\mu_i$ in this setting is

$$\mathbb{P}\left( \hat{\mu}_i^\tau \geq \mu_i + \sqrt{\frac{\log \delta^{-1}}{8N^2\tau}} \right) \leq \delta,$$

$$\mathbb{P}\left( \hat{\mu}_i^\tau \leq \mu_i - \sqrt{\frac{\log \delta^{-1}}{8N^2\tau}} \right) \leq \delta.$$

The confidence interval $\sqrt{\frac{\log \delta^{-1}}{8N^2\tau}}$ processes better concentration performance than (6) because it has an additional factor $\frac{1}{\sqrt{N}}$. Hence, the consensus estimation performance in this setting is better than that of general federated bandit settings. Hence, we could derive more convincing lower bounds on regrets.

**Lower bound proof.** Let $\mathcal{M}$ be a set of distributions with finite means, and let $\mu : \mathcal{M} \to \mathbb{R}$ be the function that maps $P \in \mathcal{M}$ to its mean. Let $\mu_{i^\star} \in \mathbb{R}$ and $P \in \mathcal{M}$ have $\mu(P) < \mu_{i^\star}$ and define
$$d_i = d_{\inf}(P, \mu_{i^\star}, \mathcal{M}) = \inf_{P' \in \mathcal{M}} \{D(P, P') : \mu(P') > \mu_{i^\star}\},$$
where $D(P, P')$ is the relative entropy between $P$ and $P'$.

Define two reward distributions as follows
$$\nu = (P_1, \ldots, P_i, \ldots, P_K),$$
$$\nu' = (P_1, \ldots, P_i', \ldots, P_K).$$

Let all arms except arm $i$ be the same in the two distributions. For arm $i$, let $\epsilon > 0$ be an arbitrary small value such that $D(P_i, P_i') \leq d_i + \epsilon$ and $\mu(P_i') > \mu_{i^\star}$.

According to Lemma 15.1 in reference [Lattimore and Szepesvári, 2020], the divergence between $\nu$ and $\nu'$ is decomposed into

$$D(\mathbb{P}_\nu, \mathbb{P}_{\nu'}) = \sum_{k=1}^{K} \mathbb{E}[\tau_{k,j}(T)]D(P_i, P_i') \overset{(a)}{\leq} \mathbb{E}[\tau_{i,j}(T)](d_i + \epsilon),$$

where equation (a) is obtained based on $D(P_j, P_j') = 0$ if $j \neq i$.

According to Bretagnolle-Huber inequality (Theorem 14.2 in Lattimore and Szepesvári [2020]), for any event $A$, we have

$$\mathbb{P}_\mu(A) + \mathbb{P}_{\mu'}(A^c) \geq \frac{1}{2} \exp(-D(\mathbb{P}_\nu, \mathbb{P}_{\nu'})) \geq \frac{1}{2} \exp(-\mathbb{E}[\tau_{i,j}(T)](d_i + \epsilon))$$

Choose $A = \{\tau_{i,j}(T) > T/2\}$, and let $R_T = R_T(\mathcal{A}, \nu)$ and $R'_T = R'_T(\mathcal{A}, \nu')$. Then

$$
\begin{aligned}
R_T + R'_T &\geq \frac{T}{2}(\mathbb{P}_\mu(A)\Delta_i + \mathbb{P}_{\mu'}(A^c)(\mu'_i - \mu_{i^\star})) \\
&\geq \frac{T}{2}\min\{\Delta_i, \mu'_i - \mu_{i^\star}\}(\mathbb{P}_\mu(A)\Delta_i + \mathbb{P}_{\mu'}(A^c)) \\
&\geq \frac{T}{2}\min\{\Delta_i, \mu'_i - \mu_{i^\star}\}\exp(-\mathbb{E}[\tau_{i,j}(T)](d_i + \epsilon)).
\end{aligned}
$$

Rearranging and taking the limit inferior leads to

$$
\begin{aligned}
\liminf_{T \to \infty} \frac{\mathbb{E}[\tau_{i,j}(T)]}{\log T} &\geq \frac{1}{d_i + \epsilon}\liminf_{T \to \infty}\frac{\log\frac{T\min\{\Delta_i, \mu'_i - \mu_{i^\star}\}}{4(R_T + R'_T)}}{\log T} \\
&\geq \frac{1}{d_i + \epsilon}(1 - \liminf_{T \to \infty}\frac{\log(R_T + R'_T)}{\log T}) \\
&= \frac{1}{d_i + \epsilon},
\end{aligned}
$$

where the last equality follows from the definition of consistency, which says that for any $p > 0$, there exists a constant $C_p$ such that for sufficiently large $T$, $R_T + R'_T \leq C_p T^p$, which implies that

$$
\liminf_{T \to \infty}\frac{\log(R_T + R'_T)}{\log T} \leq p.
$$

Considering $p > 0$ was arbitrary and $\epsilon > 0$ is limited to zero, we have

$$
\liminf_{T \to \infty}\frac{\mathbb{E}[\tau_{i,j}(T)]}{\log T} \geq \frac{1}{d_i}.
$$

In the proof of the lower bound regret, agents only use the information from their replica subsystems and the reward mean is a $\frac{1}{\sqrt{N}}$-sub-Gaussian variable. According to Table 16.1 given in Lattimore and Szepesvári [2020], we have $d_i = \frac{N\Delta_i^2}{2}$. The individual regret of the problem is lower bounded by

$$
\liminf_{T \to \infty}\frac{R_j(T)}{\log T} \geq \liminf_{T \to \infty}\sum_{i:\Delta_i > 0}\frac{\Delta_i \mathbb{E}[\tau_{i,j}(T)]}{\log T} \geq \sum_{i:\Delta_i > 0}\frac{\Delta_i}{d_i} \geq \sum_{i:\Delta_i > 0}\frac{2}{N\Delta_i}.
$$

The group lower regret is

$$
\begin{aligned}
\liminf_{T \to \infty}\frac{R(T)}{\log T} &= \liminf_{T \to \infty}\frac{\sum_{j=1}^N R_j(T)}{\log T} \geq \liminf_{T \to \infty}\sum_{j=1}^N\sum_{i:\Delta_i > 0}\frac{\Delta_i \mathbb{E}[\tau_{i,j}(T)]}{\log T} \\
&\geq \sum_{j=1}^N\sum_{i:\Delta_i > 0}\frac{\Delta_i}{d_i} \geq \sum_{i:\Delta_i > 0}\frac{2}{\Delta_i}.
\end{aligned}
$$

$\square$

# E    Appendix / Experiments on homogeneous settings

Under the conditions in Section 5, some experiments are conducted in this section. Our algorithm (EpoInc-SE) can also obtain a similar result compared to the optimal homogeneous bandit algorithm in Figure 2.

# F    Appendix / Experiments on large-scale multi-agent systems

For large-scale deployments, the agent number $N$ would play an important role in communication cost and regrets. In order to further verify the performance of our algorithm, we conducted an experiment involving 100 agents. The main results are shown in Figure 3. The proposed algorithm (EpoInc-SE) also processes the advantage in communication cost, while it can obtain a near-optimal regret compared with other algorithms.

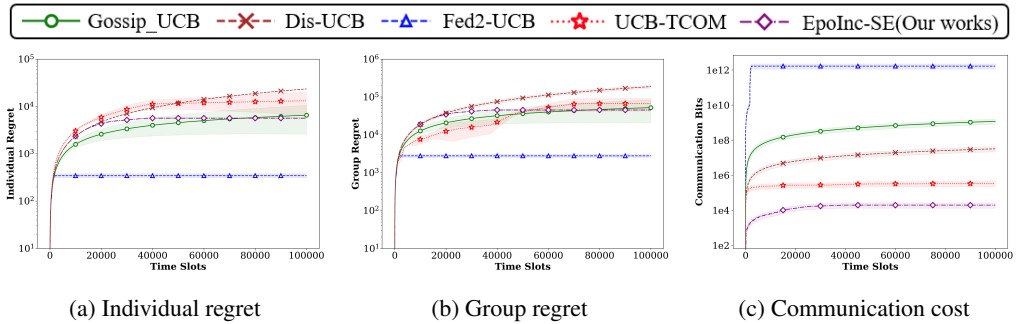

(a) Individual regret      (b) Group regret      (c) Communication cost

Figure 2: Performance in homogeneous settings.

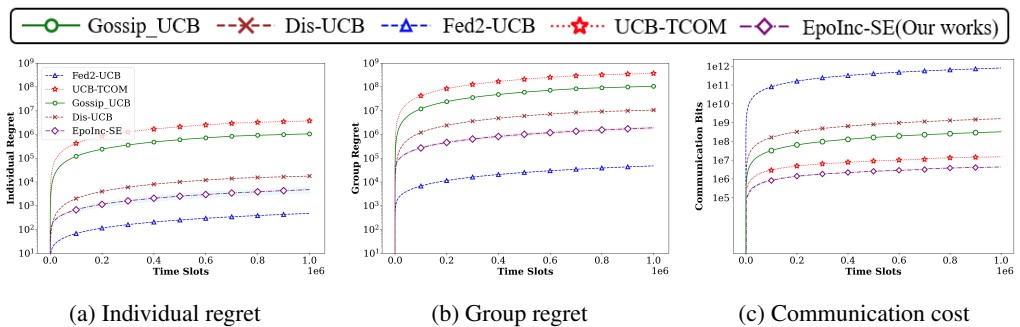

(a) Individual regret      (b) Group regret      (c) Communication cost

Figure 3: Performance in large-scale multi-agent systems.

# G    Appendix / Limitations

Although our algorithm has obtained a near-optimal result, it still has certain limitations as follows:

    a   The algorithm proposed in this work is the successive elimination algorithm, which relies on round-robin sampling. A similar result can not be introduced to UCB-based algorithms, which are the mainstream algorithms in bandit problems.

    b   For time-variant communication graphs, `EpoInc-SE` would be not work because the consensus estimation part `EBCES` could not work in random graphs. This is because Lemma 2 relies on a Markov Decision process, which needs a fixed graph.

    c   In some practical distributed systems, information may be unavailable or costly to obtain. For the first condition, the information is unavailable to some agents, which implies that the graph $\mathcal{G}$ is disconnected with a clique-connected component. Then, agents cannot learn global estimates without bias because agents lack other agents' local observations. Thus, the regret would be linear $\Omega(T)$, which has been discussed in reference [Xu and Klabjan, 2023]. For the second condition, the communication cost cannot be upper-bounded because the cost of each message is unknown.

