# OpenReview forum: "Federated Multi-armed Bandits with Efficient Bit-Level Communications"
_NeurIPS.cc/2025/Conference — NeurIPS 2025 poster_

### Official Review · Reviewer_Ftja · 2025-06-12

**Clarity:** 1
**Significance:** 2
**Originality:** 3
**Rating:** 3
**Confidence:** 2

**Summary:**

This paper studies the federated multi-armed bandit (FMAB) problem where N agents connected via a communication graph collaborate to identify the optimal arm while minimizing communication costs. The paper focuses on the heterogeneous setting where local reward distributions differ across agents. The main contribution is EpoInc-SE (Incremental Epoch-based Successive Elimination), an algorithm that combines epoch-based exploration, two-phase communication, and adaptive differential encoding to achieve near-optimal regret bounds while reducing communication to $O(\sqrt{log T})$ total bits with $O(1)$ bits per message in expectation. The authors provide both upper and lower bounds for individual and group regret, claiming their algorithm is near-optimal.

**Questions:**

1. what is the computational complexity?
2. what are the implications of these different communication schemes on privacy?
3. What is the worst-case number of bits required per message?

**Ethical Concerns:**

["NO or VERY MINOR ethics concerns only"]

**Final Justification:**

After reading the rebuttal and other reviews, my overall assessment remains that the paper is not ready for acceptance in its current form. While the authors have made a commendable effort to clarify their contributions and address specific points (e.g., separating elimination/estimation communication, adding a worst-case bit count remark, acknowledging the fairness citation), the paper still needs major revision to both presentation and  technical content. In particular:
- The manuscript needs a substantial rewrite to integrate the explanations provided in the rebuttal, improving both presentation clarity and the focus of the contributions.
- The paper still mixes per-message $\mathrm{O}(1)$ bits/message (in expectation), table-level $\mathrm{O}(\sqrt{ \log \mathrm{T})}$ vs. theorem-level $O\left(\sum_{i: \Delta_i>0} \sqrt{N} \Delta_i^{-1} \sqrt{\log T}\right)$. The worst-case bound now stated in the rebuttal is useful, but the paper still lacks a fully specified, reproducible message schema and unified metric. This undermines one of the main selling points, as also noted by Reviewer Md65.
- Other tecchnical consistency issue (such as the global reward definition confusion noted by Reviewer Md65) also requires more careful polishing.
Given these points, my opinion remains unchanged, and I lean toward rejection in its current form.

**Limitations:**

Yes, they provided discussions to some limitations. More on computational complexity and privacy implications should also be discussed

**Paper Formatting Concerns:**

N.A.

**Quality:**

3

**Strengths And Weaknesses:**

# Strengths:

- The paper addresses a practically relevant setting combining federated learning with communication efficiency, which is crucial for distributed systems with bandwidth constraints.
- The adaptive differential encoding technique that reduces message size from $O(\sqrt{log T})$ to $O(1)$ bits (in expectation) is creative and represents a genuine technical contribution to the federated bandits literature, to the best of my knowledge.
- The separation of urgent information (arm elimination) from routine updates (estimates) is an elegant design principle that balances latency and efficiency.

# weakness

1. The paper could benefit from a clearer articulation of the precise problem it aims to solve. Currently, it addresses multiple aspects—communication efficiency, heterogeneity handling, optimal regret, and consensus estimation—without clearly identifying which is the primary contribution versus supporting components. This scattered presentation makes it challenging to evaluate the paper’s specific goals and their contribution to the existing literature. To strengthen the work, the authors are encouraged to explicitly state the primary problem, separate necessary technical elements from the key scholarly advancements, and include a comparison of communication costs with prior methods to highlight the approach’s improvements.
2. in lines 40-42, Srikant & Ying study service‐fairness in queueing networks—not learning agents’ regret—so the citation is a domain mismatch. And the statement itself is not rigorous as "fairness" has been formalized in several distinct ways in the bandits literature.
3. The O(1) bit communication claim is only in expectation, not worst-case, which is not clearly stated in the main results.
4. The lower bound proof uses "replica subsystems" that fundamentally change the problem by giving each agent access to N times more information, making the comparison invalid.
5. Experiments only use synthetic data with small problem instances (N=8, K=10). And there is no direct validation of the O(1) bit communication claim. The authors are encouraged to conduct ablation studies as the work involves multiple components.

Minor

- multiple grammar issues and typos (e.g., "were" vs “are” in line 50; "with non-server" in line 51; a space missing in line 52, etc.)

---

> ### Author Rebuttal · Authors · 2025-07-31
>
> # On contribution
> >Q: The paper could benefit from a clearer articulation of the precise problem it aims to solve. Currently, it addresses multiple aspects—communication efficiency, heterogeneity handling, optimal regret, and consensus estimation—without clearly identifying which is the primary contribution versus supporting components. This scattered presentation makes it challenging to evaluate the paper’s specific goals and their contribution to the existing literature. To strengthen the work, the authors are encouraged to explicitly state the primary problem, separate necessary technical elements from the key scholarly advancements, and include a comparison of communication costs with prior methods to highlight the approach’s improvements.
>
> A: Thank the reviewer for the comment. The core goal of this article is to optimize the communication cost while maintaining the near-optimal regret. The consensus estimation is to handle the heterogeneous feedback, and the optimal regret is an intermediate result. For the novelty, we list them as follows
>
> **(1) Adaptively Quantized communication for arm elimination:** The first novelty is that transmitting information precisely meets the requirement of arm elimination at the current moment. In the communication optimization problem, the most important challenge is to avoid the waste caused by the premature transmission of precise information. To handle the challenge, we design an adaptive difference communication policy to reduce the message size per round.
>
> **(2) Suitable epoch design for unbiased estimation:** The second novelty is to design a suitable epoch-based exploration policy that matches the consensus estimation policy, to estimate the global mean without bias. In the consensus estimation policy, the information from the current epoch typically needs to be integrated with historical information. Both excessively large and overly small epoch sizes can introduce bias in the consensus estimation policy. To ensure unbiased estimation, we adopt a novel epoch size to avoid excessive accumulation of sampling information within the current epoch.
>
> **(3) Two-phase communication strategy for reducing time delay:** The third novelty is in designing two separate communication pathways to reduce the time delay caused by the fully distributed graph. For the general information (global estimates), it broadcasts at the end of the epoch. For urgent information (eliminated arms), immediate informing all agents is needed. Hence, we introduce the other communication phase to broadcast them, which reduces the time delay from $D$ epochs to one or two epochs.
>
> For the lower bound, while the lower bound proofs of our Theorem 3 and that in [1] both follow the standard three steps---(1) Construct the hard instances, (2) Devise a quantity to measure the difference/distance between the instances. (3) Use an asymptotic KL-divergence argument to prove the lower bound, the steps (1) and (2) in our federated bandit rely on different techniques from those of [1]. Step (1) carefully chooses the instances different for *each* agent, and Step (2) measures the concentration difference in terms of our consensus communication, both of which are different from [1].
>
> # On literature
>
> >Q: In lines 40-42, Srikant & Ying study service‐fairness in queueing networks—not learning agents’ regret—so the citation is a domain mismatch. And the statement itself is not rigorous as "fairness" has been formalized in several distinct ways in the bandits literature.
>
> A: Thank the reviewer for pointing out this issue. We mainly use the definition of max-min fairness in Section 2.1.1, but not the full book. Hence, we will modify our clarity of this part as follows
>
> "In network optimization literature [1], the max-min fairness metric—maximize the minimal individual reward—is widely used to measure a system’s fairness. "
>
> # On communication cost
>
> >Q: The $O(1)$ bit communication claim is only in expectation, not worst-case, which is not clearly stated in the main results.
>
> >Q: What is the worst-case number of bits required per message?
>
> A: Thank the reviewer for the insightful suggestion. We will add a remark after Theorem 2 to explain the bits consumed at each time slot as follows
>
> "**Remark 3**: In the worst case, the message in the communication phase I will consume at most $KN(2+\hat{l})$ bits. In the communication phase II, the message will consume at most $K\log K+K\log\log T$ bits."
>
> >Q: Experiments only use synthetic data with small problem instances (N=8, K=10). And there is no direct validation of the $O(1)$ bit communication claim. The authors are encouraged to conduct ablation studies as the work involves multiple components.
>
> A: Thank the reviewer for the comment. We have added the experiment on large-scale multi-agent systems. The result is shown in the following table, which is in line with our theoretical proof. For the ablation studies, we will add them in the latest version.
>
> | Name   |  Individual regret     | Group regret   | Communication cost |
> |:--------|:------------:|---------:|---------:|
> | Fed2-UCB   | $10^4$     | $10^6 $   |  $10^9$  |
> | Dis-UCB  | $10^6$     | $10^8$    |  $10^8$  |
> | EpoInc-SE  | $10^5$     | $10^7$    |  $10^6$  |
>
> # On the proof of the lower bound
>
> >Q: The lower bound proof uses "replica subsystems" that fundamentally change the problem by giving each agent access to N times more information, making the comparison invalid.
>
> A: Thank the reviewer for the comment. In Theorem 3, we are focused on searching for a lower bound. More information gives each agent access to N times more information, and the lower bound of this problem will be lower than that of the original problem. To find the lower bound of the original problem is different, while we can consider the new problem's lower bound as the lower bound of our problem.
>
> # On computational complexity
>
> >Q: What is the computational complexity?
>
> A: Thank the reviewer for the valuable comment. The computational complexity of our algorithm is $O(\log TN^2K)$.
>
> # On privacy
>
> >Q: What are the implications of these different communication schemes on privacy?
>
> A: Thank the reviewer for the comment. The privacy of users is protected in our algorithm because we only transmit the differential quantization value of the global estimates, while the practical rewards $X_{i,j}(t)$ are not transformed, which greatly protects the privacy of statistics. However, in some algorithms, the reward  $X_{i,j}(t)$ is directly transmitted between agents, which is easily attacked.
>
> # On writing
>
> >Q: Multiple grammar issues and typos (e.g., "were" vs “are” in line 50; "with non-server" in line 51; a space missing in line 52, etc.)
>
> A: Thank the reviewer for the correction. We will fix the typo in the latest revision.
>
>
> # Reference
>
> [1] Rayadurgam Srikant and Lei Ying. Communication networks: An optimization, control and stochastic networks perspective. Cambridge University Press, 2014.

---

> > ### Comment · Reviewer_Ftja · 2025-08-02
> >
> > I thank the authors for their detailed rebuttal, which largely addresses my concerns. I will wait after discussing with other reviewers and the AC to see if I should adjust my overall evaluation. Good luck.

---

> > > ### Author Response · Authors · 2025-08-04
> > >
> > > Thank you for your prompt follow-up. We will incorporate these improvements into the next revision.

---

### Official Review · Reviewer_3qLe · 2025-06-21

**Clarity:** 3
**Significance:** 2
**Originality:** 2
**Rating:** 4
**Confidence:** 3

**Summary:**

This paper addresses the Federated Multi-Armed Bandit (FMAB) problem. Under this problem setting, multiple agents aim to minimize cumulative regret while constrained by limited communication cost and privacy restrictions. In contrast to centralized bandit models, FMAB agents operate in a decentralized network, communicating only with neighbors, which makes effective coordination challenging. The paper proposes a new algorithm, EpoInc-SE (Incremental Epoch-based Successive Elimination), which is designed to balance communication cost with learning performance by incorporating early arm elimination, epoch-based exploration, and a quantized, differentially transmitted statistics mechanism. The paper shows that the algorithm achieves near-optimal regret both at the group and individual levels, and matches the theoretical lower bounds on regret in heterogeneous federated settings. EpoInc-SE is the first method to consider communication bit cost in such settings, introducing a two-phase communication strategy to prioritize important information and reduce overhead. Additionally, the authors propose a consensus estimator subroutine (EBCES) that allows agents to estimate global arm rewards accurately under distributed communication constraints. Theoretical analysis and empirical evaluations show the algorithm’s effectiveness, demonstrating good performance in both regret minimization and communication efficiency compared to existing approaches.

**Questions:**

1. What is the notable technical challenge in the theoretical analysis of the upper and/or lower bounds?

2. The experiments are run on a small network (i.e., 8 agents). Any comment on how communication and regret behave in more realistic, large-scale deployments?

**Ethical Concerns:**

["NO or VERY MINOR ethics concerns only"]

**Limitations:**

Yes

**Quality:**

3

**Strengths And Weaknesses:**

**Strengths**

**Quality**

The paper has a fine quality. The proposed algorithm, EpoInc-SE, is well-motivated and specifically designed to deal with the communication efficiency in federated bandit. The theoretical analysis covers both regret upper and lower bounds, and shows that the algorithm is near-optimal.

**Clarity**

The paper is largely well-structured. On the theoretical part, mathematical concepts and definitions are well introduced. Section 3 presents a good introduction of the propose algorithm.

**Significance**

The work addresses a meaningful and increasingly important problem—efficient distributed learning in the presence of communication constraints and heterogeneity. Federated bandit and distributed decision-making is an important research topic and applicable to various areas such as finance and network optimization. The paper's theoretical contribution is a good one. The practical implications of reducing communication to O(1) bits per message while preserving regret performance are substantial.

**Originality**

The paper introduces several novel elements. It is, to the authors’ knowledge, the first to optimize bit-level communication in heterogeneous FMAB settings and to propose a lower bound on regret that explicitly incorporates agent heterogeneity.

**Weaknesses**

**Lack of Novel Analysis Techniques**

The theoretical analysis relies entirely on standard techniques like Hoeffding’s inequality, spectral graph theory, and existing lower bound arguments from prior work. It does not introduce new analytical tools or proof techniques. The regret decomposition into quantization, consensus, and concentration errors is standard, and the lower bounds are derived using known results in a relaxed model. For a theory-focused paper, this lack of technical novelty in the analysis is a notable weakness.

**Minor Weakness**

The empirical evaluation is limited to synthetic experiments; performance on real-world federated datasets would make the results more compelling.

---

> ### Author Rebuttal · Authors · 2025-07-31
>
> # On novelty
>
> >Q: The theoretical analysis relies entirely on standard techniques like Hoeffding’s inequality, spectral graph theory, and existing lower bound arguments from prior work. It does not introduce new analytical tools or proof techniques. The regret decomposition into quantization, consensus, and concentration errors is standard, and the lower bounds are derived using known results in a relaxed model. For a theory-focused paper, this lack of technical novelty in the analysis is a notable weakness.
>
> A:  Thank the reviewer for the comment. Our work is indeed built upon the three previously mentioned works. However, the existing results cannot be obtained through a mere aggregation of prior work without additional innovation. We highlight three novel contributions of our work:
>
> **Adaptively Quantized communication for arm elimination:** The first novelty is that transmitting information precisely meets the requirement of arm elimination at the current moment. In the communication optimization problem, the most important challenge is to avoid the waste caused by the premature transmission of precise information. To handle the challenge, we design an adaptive difference communication policy (Section 3.3) to reduce the message size per round.
>
> **Suitable epoch design for unbiased estimation:** The second novelty is to design a suitable epoch-based exploration policy that matches the consensus estimation policy, so as to estimate the global mean without bias. In the consensus estimation policy (Section 3.4), the information from the current epoch typically needs to be integrated with historical information. Both excessively large and overly small epoch sizes can introduce bias in the consensus estimation policy. To avoid the biased consensus estimation, we adopt an accumulative epoch size instead of the traditional binary epoch size to avoid excessive accumulation of sampling information within the current epoch.
>
> **Two-phase communication strategy for reducing time delay:** The third novelty is in designing two separate communication pathways to reduce the time delay caused by the fully distributed graph. For the general information (global estimates), it broadcasts at the end of the epoch. For the urgent information (eliminated arms), we need to immediately inform all the agents. Hence, we introduce the other communication phase to broadcast them, which reduces the time delay from $D$ epochs to one or two epochs.
>
> For the lower bound proof, while the lower bound proofs of our Theorem 3 and that in [1] both follow the standard three steps---(1) Construct the hard instances, (2) Devise a quantity to measure the difference/distance between the instances. (3) Use an asymptotic KL-divergence argument to prove the lower bound; the steps (1) and (2) in our federated bandit rely on different techniques from those of [1]. Step (1) carefully chooses the instances different for *each* agent, and Step (2) measures the concentration difference in terms of our consensus communication, both of which are different from [1].
>
> # On experiment
>
> >Q: The empirical evaluation is limited to synthetic experiments; performance on real-world federated datasets would make the results more compelling.
>
> A: Thank the reviewer for the valuable comment. We will add real-world federated datasets in the latest version.
>
> >Q: The experiments are run on a small network (i.e., 8 agents). Any comment on how communication and regret behave in more realistic, large-scale deployments?
>
> A: Thank you for your review. For large-scale deployments, the agent number $N$ would play an important role in communication cost and the individual regret. In order to further verify the performance of our algorithm, we conducted an experiment involving 100 agents. Due to the limitations of the format, we present the performance results of the three algorithms in the large-scale multi-agent system in the form of a table here. From the table, we can learn that our algorithm consumes the least resources and obtains a near-optimal result.
>
> | Name   |  Individual regret     | Group regret   | Communication cost |
> |:--------|:------------:|---------:|---------:|
> | Fed2-UCB   | $10^4 $    | $10^6 $  | $ 10^9 $ |
> | Dis-UCB  | $10^6$     | $10^8 $   |  $10^8$  |
> | EpoInc-SE  | $10^5 $    | $10^7$    |  $10^6$  |
>
>
>
>
> # Reference
>
> [1] Lattimore, Tor, and Csaba Szepesvári. Bandit algorithms. Cambridge University Press, 2020.

---

> ### Comment · Reviewer_3qLe · 2025-08-04
> **Response to rebuttal**
>
> I thank the authors for their effort in the rebuttal, which addressed my questions. I also read the comments from other reviewers and the corresponding response from the authors. I will take all the information into consideration during the discussion with other reviewers. Thank you!

---

> > ### Author Response · Authors · 2025-08-04
> >
> > Thank you for your prompt follow-up. We will incorporate these improvements into the revision.

---

### Official Review · Reviewer_7iqF · 2025-06-26

**Clarity:** 4
**Significance:** 3
**Originality:** 4
**Rating:** 5
**Confidence:** 3

**Summary:**

The authors consider the problem of cumulative regret minimization in decentralized heterogeneous federated bandits with non-complete communication graphs. Agents can only communicate with their neighbors on the graph and aim to find the optimal arm across agents while having access to potentially diverse local expected rewards. They propose an algorithm called EpoInc-SE which features several aspects: epoch-based increasing uniform sampling of arms, modulated communication with distinct phases depending on the type of transmitted information, computation of quantized estimates and transmission of differential estimates across neighbors to reduce the communication cost, and refined elimination criterion to take into account quantization and consensus estimation errors in addition to the standard concentration inequality. They show that their algorithmic contribution has an improved upper bound on both the individual and group regrets compared to the state-of-the-art in heterogeneous bandits and asymptotically matching the lower bound, while considerably reducing the communication cost. Finally, experiments are performed on synthetic data compared to the baselines in heterogeneous and homogeneous, centralized and decentralized bandits.

**Questions:**

- Can you discuss the importance and the source of the agents’ weights in the communication graph and its potential impact on the regret upper bounds?

Minor concern: Ideally, the code should feature a README file to document which commands should be run for each figure in the paper.

**Ethical Concerns:**

["NO or VERY MINOR ethics concerns only"]

**Final Justification:**

The part about the communication graph is solved in the rebuttal to Md65 and myself I think. Albeit it is a strong assumption in practice, similar assumptions are present in the literature, so I think the paper's theoretical work is still worthwhile. Combining the three aspects (epoch-based increasing uniform sampling of arms, modulated communication, computation of quantized estimates and transmission of differential estimates, refined elimination criterion) does not seem straightforward, so I would think that the significance of the work is good enough, and possibly usable in related problems.

There are indeed some presentation issues as highlighted by Ftja, but they seem solvable within the camera-ready deadline. The paper still brings an interesting contribution to the topic with solid technical elements, so I would lean towards acceptance.

**Limitations:**

Yes.

**Paper Formatting Concerns:**

None.

**Quality:**

4

**Strengths And Weaknesses:**

**Strengths:**
- Quality: The paper looks technically solid. The limitations mentioned in the paper make sense but should perhaps be featured in the main paper instead of the appendix.
- Clarity: The paper is well-written, and its algorithmic and theoretical contributions are clear.
- Significance: The proposed algorithm matches the lower bound for cumulative individual and group regret minimization (which the authors rederive), while considerably reducing the upper bound on the communication cost. These results seem confirmed by the experimental results in Section 5. In particular, this is the first algorithm for heterogeneous federated bandits matching those bounds.
- Originality: The quantization and consensus algorithm for estimation are interesting, rather new contributions which required refined confidence bounds for the arm elimination procedure.

**Weaknesses:**
- Significance: The importance and the source of the agents’ weights in the communication graph is difficult to assess. Reading the appendix (Section B), it seems that this matrix is roughly fixed, up to a parameter $\beta$ which exact value in practice is unspecified. I think this information should be present in the main text.

---

> ### Author Rebuttal · Authors · 2025-07-31
>
> # On graph
>
> >Q: Significance: The importance and the source of the agents’ weights in the communication graph is difficult to assess. Reading the appendix (Section B), it seems that this matrix is roughly fixed, up to a parameter $\beta$ which exact value in practice is unspecified. I think this information should be present in the main text.
>
> A: Thank the reviewer for the insightful suggestion. We will add the definition of $W$ and $\beta$ in Line 104, Section 2. The concrete content is as follows
>
> "The definition of $W$ is $W=I−\beta\mathcal{L}$, where $\mathcal{L}$ is the Laplacian matrix and $\beta\in(0, 1/N]$ is the coefficient that reflects whether this agent believes its neighbors. If $\beta=0$, we have $W=I$, which means that the agent itself occupies all the weight. If$\beta=1/N$, it indicates that the agent assigns equal trust to all of its neighbors."
>
> >Q: Can you discuss the importance and the source of the agents’ weights in the communication graph and its potential impact on the regret upper bounds?
>
> A: Thank the reviewer for the comment. The agent weights determine how each agent integrates information from its neighbors. The weights between agents determine the connectivity of information flow, which in turn affects the convergence of the consensus estimation. Higher weights accelerate the spread of local observations or estimates across the network.  Properly balanced weights help ensure all agents converge to a common estimate or belief. The details have been written in Appendix B.
>
> The weight matrix influences the convergence of consensus estimation, which could lead to additional constant-level regret because more sample counts are needed. The details of regrets are shown in the proof of Theorem 1.
>
> # On README
>
> >Q: Minor concern: Ideally, the code should feature a README file to document which commands should be run for each figure in the paper.
>
> A: Thank you for your suggestion. We will add a README file in the new version to explain how to run each figure in this paper.

---

> ### Comment · Reviewer_7iqF · 2025-08-04
>
> Thank you for addressing my comments. Reviewer Md65's question on the communication graph and the subsequent response addressed my additional concern.  I will wait for the discussion phase with other reviewers and the AC to confirm my opinion of keeping the score as it is.

---

> ### Author Response · Authors · 2025-08-04
>
> Thank you for your prompt follow-up. We will improve the points above and details in the future work.

---

### Official Review · Reviewer_Md65 · 2025-06-30

**Clarity:** 2
**Significance:** 2
**Originality:** 2
**Rating:** 5
**Confidence:** 4

**Summary:**

This paper addresses the Federated Multi-Armed Bandit (FMAB) problem in fully distributed networks with communication constraints. The main contribution is EpoInc-SE, a communication-efficient successive elimination algorithm that balances regret minimization with reduced bit-level communication. The algorithm incorporates (1) an epoch-based exploration strategy, (2) an adaptive differential communication scheme that transmits only quantized incremental updates, and (3) a consensus estimation routine to aggregate local statistics. Theoretically, the paper establishes tight upper and lower bounds on both individual and group regret, showing that EpoInc-SE achieves near-optimal performance. Empirically, it outperforms or matches baselines in regret while significantly reducing communication cost.

**Questions:**

1. Given my concerns regarding the novelty of the work, could you clarify the main technical challenges involved in proving the regret and communication upper bounds presented in Theorems 1 and 2?
2. Additionally, in Theorem 3, what distinguishes your proof of the regret lower bounds from the arguments of a similar lower bound for the single-agent case in [1], which your approach seems to closely follow? What are the key technical hurdles specific to your setting?
3. What are the main obstacles to establishing a lower bound for the communication cost? Do you believe the current upper bound of $O(\sqrt{\log T})$ is tight, or is there potential for further improvement?
4. In Section 5, you mention incorporating a consensus estimation module into Fed2-UCB and UCB-TCOM, but the details are not elaborated. Could you provide more information on this?
5. In lines 111–112, the global rewards are defined as sums over agents' local rewards, whereas the global means are defined as averages of the agents’ local means. Is this intentional?

[1] Lattimore, Tor, and Csaba Szepesvári. Bandit algorithms. Cambridge University Press, 2020.

**Ethical Concerns:**

["NO or VERY MINOR ethics concerns only"]

**Final Justification:**

My comment to the AC details my reasoning for changing the score.

**Limitations:**

Yes

**Paper Formatting Concerns:**

I did not notice any major formatting issues.

**Quality:**

3

**Strengths And Weaknesses:**

**Strengths:**
- **Quality:** The paper’s main claims are supported by rigorous and well-structured proofs, demonstrating a solid theoretical foundation. Additionally, the paper includes a basic experimental evaluation that demonstrates the superiority of the proposed methods over existing baselines.
- **Clarity:** The proposed algorithm and accompanying proof techniques are presented with intuitive explanations in the main text, and their effectiveness is validated through a basic experimental evaluation.
- **Significance:** This work advances the state of the art by introducing a distributed algorithm for the heterogeneous FMAB setting with the lowest known communication cost, and with tight individual and group regret upper bounds.
- **Originality:** Although the authors build on established techniques such as quantized communication [1], epoch-based communication [2] and successive elimination-based implicit synchronization [3], they employ them in a novel manner in the heterogeneous setting to achieve an algorithm with tight bounds and low communication cost.

**Weaknesses:**
Rather than separating into broad quality, clarity, significance and originality categories, I will outline my main concerns in a more detailed manner below.
- My primary concern with this work lies in its level of novelty. The contributions appear to be incremental, building upon established techniques in cooperative bandit literature—such as epoch-based communication [2], successive elimination with implicit synchronization [3], and running-consensus estimation [4]. That said, my current score reflects the view that this is a complete study in a new heterogeneous setting. It rigorously analyzes both group and individual regret under communication constraints, matches known lower bounds, and includes a basic experimental validation.
- Implementing the proposed algorithm appears to require agents to have access to global properties of the communication graph, such as its diameter, the eigenvalues of the weight matrix, and whether the graph is balanced. This raises concerns regarding the practicality of the approach, particularly in fully distributed settings where such information may be unavailable or costly to obtain. These limitations should be explicitly discussed in the paper.
- Furthermore, the analysis of the proposed algorithm relies heavily on implicit synchronization among agents when selecting arms. This dependence raises additional concerns about the algorithm’s applicability in more realistic settings, such as those involving unreliable communication or asynchronous decision-making.

Other issues –
- The authors highlight low communication cost as a key contribution of their work. However, it remains unclear whether the stated $O(\sqrt{\log T})$ bit complexity is optimal, as the paper does not provide a corresponding lower bound. Including such a bound or discussing the potential for further improvement would strengthen the contribution.
- While the intuitive explanation of the algorithm in Section 3 is helpful, the use of numerous similar notations in this section makes the presentation difficult to follow. Additionally, Section 2 would benefit from revision to improve clarity.
- Unfortunately, the main text does not include proof outlines for Theorems 1, 2, and 3. Given that this is primarily a theoretical contribution, providing such outlines is important for guiding readers through the core ideas of the analysis. I strongly recommend including them in a revised version of the paper.
- Additional proofreading would improve the paper. For example, the meaning of the parameter $a$ should be explained at its first appearance (line 196). In line 197, “epochs” should be used instead of “epoch.” Referring to the global estimator in line 203 before its formal definition is confusing and should be avoided. In equation (5), the quantized estimators of neighbors should be used instead of the true estimators, which are not available to individual agents. Line 260 describes a sample count bound, but Lemma 1 provides a concentration bound for the means. Lines 266–267 contain unnecessary repetition, and the reference to Section 3 in line 300 appears to be misplaced. Finally, in line 305, the notation “C” should be used instead of “c”.

[1] Shi, Chengshuai, et al. "Heterogeneous multi-player multi-armed bandits: Closing the gap and generalization." Advances in neural information processing systems 34 (2021): 22392-22404.\
[2] Wang, Yuanhao, et al. "Distributed bandit learning: Near-optimal regret with efficient communication." arXiv preprint arXiv:1904.06309 (2019).\
[3] Yang, Lin, et al. "Cooperative multi-agent bandits: Distributed algorithms with optimal individual regret and constant communication costs." arXiv preprint arXiv:2308.04314 (2023).\
[4] Martínez-Rubio, David, Varun Kanade, and Patrick Rebeschini. "Decentralized cooperative stochastic bandits." Advances in Neural Information Processing Systems 32 (2019).

---

> ### Author Rebuttal · Authors · 2025-07-31
>
> # On novelty and challenge
> >Q: My primary concern with this work lies in its level of novelty ....
>
> A: We sincerely appreciate your clear and succinct summary of our contribution. Our work is indeed built upon the three previously mentioned works. However, the existing results cannot be obtained through a mere aggregation of prior work without additional innovation. We highlight three novel contributions of our work:
>
> **(1) Adaptively Quantized communication for arm elimination:** The first novelty is that transmitting information precisely meets the requirement of arm elimination at the current moment. In the communication optimization problem, the most important challenge is to avoid the waste caused by the premature transmission of precise information. To handle the challenge, we design an adaptive difference communication policy to reduce the message size per round.
>
> **(2) Suitable epoch design for unbiased estimation:** The second novelty is to design a suitable epoch-based exploration policy that matches the consensus estimation policy, to estimate the global mean without bias. In the consensus estimation policy, the information from the current epoch typically needs to be integrated with historical information. Both excessively large and overly small epoch sizes can introduce bias in the consensus estimation policy. To ensure unbiased estimation, we adopt a novel epoch size to avoid excessive accumulation of sampling information within the current epoch.
>
> **(3) Two-phase communication strategy for reducing time delay:** The third novelty is in designing two separate communication pathways to reduce the time delay caused by the fully distributed graph. For the general information (global estimates), it broadcasts at the end of the epoch. For urgent information (eliminated arms), immediate informing all agents is needed. Hence, we introduce the other communication phase to broadcast them, which reduces the time delay from $D$ epochs to one or two epochs.
>
> >Q: Given my concerns regarding the novelty of the work, could you clarify the main technical challenges involved in proving the regret and communication upper bounds presented in Theorems 1 and 2?
>
> A: Thank the reviewer for the comment. Our main technical challenge lies in how to reduce the number of communication bits, while maintaining a near-optimal regret performance. In the most recent prior work, the number of communication rounds was reduced to $O(\log T)$. However, each of these communication rounds incurs a large number of bits cost, which can be as large as $O(T)$. While many sophisticated communication protocols proposed in the non-federated multi-agent bandits problems can be applied here to reduce both the number of communication rounds and the per-round bits, our problem setting is more challenging in three main aspects.
>
> (1) How to avoid transmitting useless information is one of the key challenges. In the process of the algorithm, the premature transmission of precise information is a waste of resources, which accounts for a major component of the communication cost.
>
> (2) How to avoid the biased estimation caused by the unreasonable epoch size. Both excessively large and overly small epoch sizes can introduce bias in the consensus estimation policy.
>
> (3) How to deal with the time delay is the third challenge. Time delay in epoch-based algorithms is fatal because the delay may occupy $O(\log T)$ rounds.
>
> # On limitations
> >Q: Implementing the proposed algorithm appears to require ...
>
> A: We thank the reviewer for the insightful comment. While the lack of global property can make our algorithms invalid, obtaining this global information is usually inexpensive.
> Given that the agents can communicate with each other and are aware of their local graph (e.g., their neighbors and their weights), a warm-up stage with merely several rounds (at most $N^2$) is enough for all agents to get the full global property.
>
> In cases where the communication is unavailable or costly, this is a very different assumption from where our paper stands. It is an interesting work that we will research in the future. Meanwhile, we will add a limitation in Appendix / Limitations
>
>  “**F Appendix / Limitations**: In practical distributed systems, information may be unavailable or costly to obtain. (a) For the first condition, the information is unavailable to some agents, which implies that the graph $\mathcal{G}$ is disconnected with a clique-connected component. Then, agents cannot learn global estimates without bias because agents lack other agents' local observations. Thus, the regret would be linear $\Omega(T)$, which has been proposed in reference [2]. (b) For the second condition, the communication cost cannot be upper bounded because the cost of each message is unknown. ”
>
> >Q: Furthermore, the analysis of the proposed algorithm...
>
> A: Thank the reviewer for the comment. Synchronized communication is the core characteristic of the Successive Elimination algorithm, which has been used in works [1,3]. Hence, our work is common practice. We agree that they are intriguing directions. Nevertheless, they are out of the scope of this paper. There are also many asynchronous algorithms, such as UCB-based algorithms [4,5]. However, the existing algorithms cannot obtain a performance similar to this work, which has been listed in Table 1. Regarding unreliable communication, it can be considered that the information is unavailable, which has been discussed in weakness 2.  To solve the problem, we try to combine UCB with SE to reduce the influence of implicit communication. We will incorporate this direction into our discussion of future work.
>
> # On the communication cost
> >Q: The authors highlight low communication cost...
>
> >Q: What are the main obstacles to establishing...
>
> Thank the reviewer for the valuable comments. There are some results on lower bounds of communication cost in homogeneous bandit problems [3], but heterogeneous and homogeneous are not the same. We will add a discussion after Theorem 2 as follows,
>
> "**Remark 2**: The lower bound of the communication cost is still an important problem for heterogeneous bandit problems. There is previous work [6] that provides a constant level lower bound for the homogeneous bandit. However, the heterogeneous bandit setting poses a more challenging challenge, i.e., heterogeneous feedback, which implies that more essential communication is needed for a near-optimal algorithm. It is an interesting work that is worth learning. "
>
> # On the proof
> >Q: Unfortunately, the main text does not include proof outlines...
>
> A: Thank the reviewer for the insightful suggestion. We will add the proof sketches in the latest version. Due to space limitations, we are unable to provide the full details here. However, a summary can be found in the subparts of the detailed proof section.
>
> >Q: Additionally, in Theorem 3, what distinguishes...
>
> A: While the lower bound proofs of our Theorem 3 and that in [7] both follow the standard three steps---(1) Construct the hard instances, (2) Devise a quantity to measure the difference/distance between the instances. (3) Use an asymptotic KL-divergence argument to prove the lower bound, the steps (1) and (2) in our federated bandit rely on different techniques from those of [1]. Step (1) carefully chooses the instances different for *each* agent, and Step (2) measures the concentration difference in terms of our consensus communication, both of which are different from [7].
>
> # On definition
> >Q: In lines 111–112, the global rewards are defined
>
> A: We thank the reviewer for pointing out this confusion. Below, we clarify the definition of the global mean of each arm. We will introduce the setting from the following two angles.
>
> (1) Application: Since each agent can only observe local sampling information, it needs to aggregate information from all other agents to evaluate the overall performance of arms, which is also a key challenge in federated bandit learning. With the loss of a central server, enabling agents to collaboratively perform information collection and aggregation through distributed communication and identifying the optimal arm is the core contribution of this paper. This topic has strong practical significance in the medical field [8].
>
> (2) Previous work: Both the definition of heterogeneity and the heterogeneous case are presented in many previous works, such as [4,5] and we follow the problem setting of them.
>
> # On clarity
> >Q: While the intuitive explanation of the...
>
> >Q: Additional proofreading would improve the paper...
>
> A: We appreciate the reviewer’s valuable feedback. For the explanation of symbols, we have proposed a detailed symbol table in Appendix A. To help with reading, we will add a refined table in Section 2. We will fix these typos in the latest revision.
>
> >Q: In Section 5, you mention incorporating a consensus estimation...
>
> A: Thank you for your correction. We have added a consensus estimation module to help them address the heterogeneity. We will add some details in the new version as follows
>
> "For Fed2-UCB and UCB-TCOM, a consensus estimation module is attached to them to help them against heterogeneity. The two algorithms communicate the historical reward at each time slot. Hence, the consensus estimator is
> $\hat{\mu}\_i(t)=\frac{\sum\_{j=1}\^N \sum\_{k=1}^t X\_{i,j}(k)}{Nt}$."
>
> # References
> [1] Optimal algorithms for multiplayer multi-armed bandits
>
> [2] Regret lower bounds in multi-agent multi-armed bandit
>
> [3] Cooperative multi-agent bandits: Distributed algorithms with optimal individual regret and communication costs
>
> [4] Distributed multi-armed bandits
>
> [5] Decentralized randomly distributed multi-agent multi-armed bandit with heterogeneous rewards
>
> [6] Distributed bandit learning: Near-optimal regret with efficient communication
>
> [7] Bandit algorithms
>
> [8] Federated learning for healthcare: Systematic review and architecture proposal

---

> > ### Comment · Reviewer_Md65 · 2025-08-04
> >
> > I thank the authors for their detailed response and efforts. I was particularly pleased to see the revised sections and the additions that will be included in the final version of the manuscript.
> >
> > My main concern with this work lies in the level of novelty. The authors' response emphasizes their unique contributions through a specific combination of previously established ideas. As I noted in my original review, I find this justification reasonable. However, I still believe that the core components of the approach—such as the use of quantization, epoch-based exploration, and the lower bound proof technique—bear strong resemblance to those in prior works.
> >
> > Additional concerns, including the assumption of knowledge about global properties of the communication graph and the requirement for implicit synchronization, have also been addressed. That said, these assumptions are likewise present in several related works.
> >
> > In light of these points, and considering the points raised by other reviewers, I think the current score is suitable and will maintain it for now. I will continue to consider all information during the reviewer-AC discussion period.

---

> > > ### Author Response · Authors · 2025-08-05
> > >
> > > Thank the reviewer for the prompt follow-up. The novelty of this work is as follows: (a) An adaptive difference communication policy is proposed that minimizes message size by transmitting only quantized information for arm elimination, avoiding premature and wasteful communication. (b) A novel epoch-based exploration policy is designed that aligns with consensus estimation to ensure unbiased global mean estimation by preventing excessive accumulation of current epoch information. (c) A two-phase communication strategy is introduced that separates urgent and general information sharing, enabling immediate broadcast of eliminated arms and reducing time delay from $D$ epochs to just one or two.
> > >
> > > What is worth mentioning is that this is the first work to account for communication cost at the bit level, and our method achieves superior round-level performance compared to existing techniques.

---

### Note · Authors · 2025-08-14

We want to thank the reviewers and AC for spending their precious time evaluating this paper.
We will ensure to address all the comments in the final version of this paper as suggested in our responses.

In summary, our work makes the following contributions: (i) an adaptive difference communication policy is presented to minimize message size while ensuring timely arm elimination; (ii) we propose an epoch-based exploration that help the consensus estimation policy to achieve unbiased global estimates; (iii) a two-phase communication strategy is proposed that significantly reduces time delay in fully distributed settings; and (iv) this is the first work that optimize the communication cost at the bit level while the regret is still near-optimal.

We have addressed all concerns raised by the reviewers, especially:

[a] For the innovations raised by Reviewers Md65 and 3qLe, we clarified the novelty and challenges over prior work, as well as the contributions above.

[b] To address the concern about experiments raised by Reviewers 3qLe and Ftja, we have additional experiments about large-scale networks (100 agents). Meanwhile, the experiment using real-world federated datasets will also be added in the new version.

[c] To enhance the readability of the proof of theorems suggested by Reviewer Md65, we provided a detailed explanation of Theorem 3 and its proof sketch (as provided in the rebuttal), which will be added in the next version.

---

### Decision · Program_Chairs · 2025-09-17

**Decision:**

Accept (poster)

**Comment:**

This paper studies the federated bandit problem considering the communication cost between agents. A phased algorithm is proposed with its guarantees on the regret and the communication bits.

For this paper various aspects are discussed between the reviewers from the clarity of the presentation to the required knowledge of the communication graph and the novelty of the communication strategy. After examining these materials, I determined to recommend acceptance considering the novelty of the new communication strategy. Though the concern on the presentation still remains, the reviewers mostly came to the opinion that it is manageable within the revision toward the final version. I agree with this opinion and expect that the author thoroughly address the presentation issues as well as other concerns raised by the reviewers in the camera-ready version.

My own minor concern is the motivation for the individual/group regret. As far as I understand, the individual regret corresponds to a hypothetical reward if the all agents would have pulled the same arm. Then, the total group rewards over agents by pulling $i^{\star}$ is no better than pulling the best arm $\arg \max_i \mu_{i,j}$ for each agent, and similarly the actual reward of each agent by pulling $i^{\star}$ is no better than pulling the best arm for each agent. I believe that this point is already well-discussed in the literature and the idea of this paper also applies to other federated settings, but I would expect that this point is discussed within the paper in more details.